

# Sensitivity of airborne radio occultation to tropospheric properties over ocean and land

Feiqin Xie[1], Loknath Adhikari[1], Jennifer S. Haase[2], Brian Murphy[3,4], Kuo-Nung Wang[5], James L. Garrison[6]

[1]Department of Physical and Environmental Sciences, Texas A&M University, Corpus Christi, Texas, USA
[2]Scripps Institution of Oceanography, University of California, San Diego, California, USA
[3]Department of Earth, Atmospheric, and Planetary Sciences, Purdue University, West Lafayette, Indiana, USA
[4] Now at: Edison State Community College, Piqua, Ohio, USA
[5]Jet Propulsion Laboratory, California Institute of Technology, Pasadena, California, USA
[6]School of Aeronautics and Astronautics, Purdue University, West Lafayette, Indiana, USA

*Correspondence to*: Feiqin Xie (Feiqin.Xie@tamucc.edu) Submitted to *Atmospheric Measurement Techniques (AMT)*

**Abstract.** Airborne radio occultation (ARO) measurements collected during a ferry flight at the end of the PRE-Depression Investigation of Cloud-systems in the Tropics (PREDICT) field campaign from the Virgin Islands to Colorado are analyzed. This long flight at ~13 km altitude provided intercomparisons of bending angle retrieval techniques over a range of environments that may have different levels of atmospheric multipath propagation interference. Two especially well-adapted radio-holographic bending angle retrieval methods, full-spectrum-inversion (FSI), and phase-matching (PM), were compared with the standard geometric-optics (GO) retrieval method. Comparison of the ARO retrievals with the near-coincident ECMWF reanalysis-interim (ERA-I) profiles shows only a small root-mean-square (RMS) refractivity difference of ~0.3 % in the drier upper troposphere from ~5 km to 13 km over both land and ocean. Both the FSI and PM methods improve the ARO retrievals in the moist lower troposphere and reduce the negative bias found in the GO retrieval due to the multipath problem. In the lowest layer of the troposphere, the ARO refractivity using FSI shows a negative bias of about –2 %. The increase of the refractivity bias occurs below 5 km over the ocean and below 3.5 km over land, corresponding to the approximate altitude of large vertical moisture gradients above the ocean and land surface, respectively. In comparisons with radiosondes, the FSI ARO soundings capture well the height of layers with sharp refractivity gradients but display a negative refractivity bias inside the boundary layer. Three spaceborne radio occultation profiles within 300 km of the flight track shows a slightly larger RMS refractivity difference of ~2 %. Analysis of the 12 ARO events that were simultaneously recorded from both the top and side-looking antennas, indicates that high precision of the ARO measurements can be achieved corresponding to an RMS difference better than 0.2 % in refractivity (or ~0.4 K). The surprisingly good quality of recordings from a very simple antenna on top of an aircraft increases the feasibility of developing an operational tropospheric sounding system.





## 1 Introduction

Radio signals from Global Navigation Satellite Systems (GNSS) can be used to sense the atmosphere during a radio occultation (RO) event, when the GNSS signals traverse progressively lower (or higher) atmospheric layers as a moving
receiver sets behind (or rises above) the Earth's limb (e.g., Kursinski et al., 1995, Rocken et al., 1997). Numerous Low Earth Orbit (LEO) satellites equipped with GNSS RO receivers have been launched since the first Global Positioning System (GPS) RO mission, the GPS/Met in 1995 (Ware et al., 1995). The spaceborne GNSS RO measurements provide high vertical resolution all-weather atmospheric soundings, which complement the conventional passive infrared and microwave sounders with their relatively low vertical resolution and high horizontal resolution, and greatly contribute to global weather
forecasting. In 2006, the launch of the six-satellite Constellation Observing System for Meteorology, Ionosphere, and Climate (COSMIC) and the GNSS Receiver for Atmospheric Sounding (GRAS) on-board MetOp began producing about 3000 daily soundings globally (Anthes et al., 2008; Luntama et al., 2008). The RO soundings were operationally assimilated into the numerical weather prediction (NWP) models at many leading weather centers and demonstrated significant impact in the upper troposphere and lower stratosphere (UTLS) (e.g., Healy and Thépaut, 2006; Cucurull and Derber, 2008). The
spaceborne RO measurements have advanced knowledge of various physical processes, including the troposphere-stratosphere exchange, gravity waves, hurricane/typhoon evolution, and planetary boundary layer (see Anthes, 2011, and references therein). However, there is relatively limited impact of RO measurements in the lower troposphere, especially on mesoscale phenomena such as severe storm forecasting and small scale processes within tropical storms. The low temporal and spatial sampling rate of spaceborne RO soundings at the regional scale (e.g., only ~1 daily profile over 400 km x 400 km
area) typically cannot capture the variation of atmospheric moisture and temperature during the lifetime of mesoscale weather phenomena. In addition, RO refractivity biases seen in the lower troposphere due to uncertainty in signal tracking (e.g., Ao et al., 2003; Beyerle et al., 2006; Ao et al., 2009; Sokolovsky et al., 2010) and the presence of ducting (e.g., Sokolovsky 2003; Xie et al., 2006; Ao et al., 2007; Xie et al., 2010) lead to degraded RO retrievals and reduced impact. The upcoming COSMIC-II mission could double or triple the number of RO soundings but would still offer a limited number of
observations over mesoscale and transient weather events.

In contrast, using GNSS receivers on-board an aircraft, dense airborne RO soundings can be collected over the target region during mesoscale and transient weather events. For a receiver within the atmosphere, the Airborne RO (ARO) technique differs from the spaceborne technique (Zuffada et al., 1999; Healy et al., 2002; Xie et al. 2008) in that the raypath through the neutral atmosphere is not symmetric with respect to the tangent point (the point of closest approach to the Earth).
In addition, the nonzero atmospheric refractivity at the receiver cannot be neglected. Therefore, the RO signals from below the local horizon must be corrected for the delay due to propagation of the signals from the aircraft altitude to the GNSS satellite above the local horizon to retrieve atmospheric properties below the receiver (e.g., Healy et al., 2002). After the precise positions of the GNSS satellite and the receiver are known, the excess phase delay due to the atmospheric refraction can be derived by calculating the difference between the measured phase and the GNSS-receiver line-of-sight (LOS)





distance. The ARO signal phase and amplitude can then be inverted to derive the atmospheric bending angle, which can be further converted to refractivity through a modified inverse Abel transformation (Healy et al., 2002; Lesne et al., 2002; Xie et al., 2008).

Early field experiments demonstrated the ARO technique by using a conventional closed-loop-tracking ARO receiver
flying at a relative low altitude of ~3 km. An exploratory flight system was tested (Yoshihara et al., 2006) but only qualitative conclusions about the performance were drawn. Xie et al. (2008) developed an end-to-end ARO simulation system based on geometric optics to describe in detail the approach, and quantified several key factors affecting the accuracy of the ARO retrievals including the aircraft velocity, in-situ refractivity measurement and the atmospheric horizontal gradient. The GNSS Instrument System for Multistatic and Occultation Sensing (GISMOS) was developed for ARO
sounding and reflection measurements (Garrison et al., 2007; Voo et al., 2009). GISMOS was tested in 2008 using the National Science Foundation (NSF) Gulfstream V (GV) research aircraft at flight altitudes of approximately 14 km over the southeastern United States (Lulich et al., 2010; Muradyan, 2009, 2012). Equipped with a dual frequency conventional GPS receiver and inertial measurement unit, GISMOS provides accurate aircraft position and velocity measurements (less than 5 mm/s) which are required for precise ARO retrieval in the lower troposphere (Muradyan et al., 2010). Haase et al. (2014)
reported the first results of the ARO measurement and retrievals from the 2010 PRE-Depression Investigation of Cloud-systems in the Tropics (PREDICT) field campaign over the equatorial Atlantic Ocean (Montgomery et al., 2012). Murphy et al. (2015) presented an assessment of the accuracy of ARO bending and refractivity retrievals from the PRECICT field campaign using data from conventional geodetic quality GPS receivers installed in GISMOS. The implementation of open-loop tracking on the ARO receiver allows high-quality ARO signal tracking deep into the moist lower troposphere where
complicated signal dynamics lead to failed signal tracking by the conventional closed-loop tracking receiver (Wang et al., 2016).  This multi-path problem caused by the large moisture variation in the lower troposphere leads to significant bias in ARO retrievals based on the geometric-optics (GO) method. With the successful development and implementation of the radio-holographic retrieval algorithms, including the full spectrum inversion (FSI, e.g., Adhikari et al. 2016) and the phase matching (PM, Wang et al., 2017), ARO retrieval quality has been significantly improved in the moist lower troposphere.

ARO simulation studies (e.g., Lense et al., 2002, Xie et al., 2008) and previous field observations (Murphy et al., 2015) have demonstrated the large impact of aircraft flight geometry (e.g., altitude, direction) on the ARO sounding density and quality. Due to the much slower motion of the aircraft (~0.25 km/sec) compared to low-earth-orbit (LEO, ~7 km/sec) satellites, it generally takes over 20 minutes for an ARO receiver to record the required data from the aircraft cruise altitude (e.g., ~10 km) down to the surface, which is much longer than the spaceborne RO sounding (1-2 minute). In addition, the
tangent points during an ARO event also drift much farther (200-300 km) than the spaceborne RO event (generally less than ~100 km). During the research flights from the PREDICT field campaign, the aircraft flew at an average altitude of ~ 14 km in "lawnmower" or square spiral patterns over the central region of deep convection associated with tropical disturbances (Haase et al., 2014). Such complicated flight patterns, i.e., changing flight altitude and direction during an ARO recording,



leads to degraded ARO signal tracking and profile quality due to signal often moving away from the maximum in the GPS antenna gain pattern (Murphy et al., 2015; Wang et al., 2016).

To better evaluate the quality of the ARO measurements and avoid the complexity of ARO signal degradation due to the varying aircraft trajectory and signal orientation during the research flights of the PREDICT field campaign, this study

focuses on the ARO measurements collecting from the ferry flight during the last leg of the campaign. During this ferry flight, the aircraft cruised northwest at nearly constant altitude (~13 km) from the field station at St. Croix in the US Virgin Islands to the UCAR facility at Broomfield, Colorado on October 2, 2010. It provided a better ARO geometry that could lead to higher quality ARO measurements. In addition, the large contrast of atmospheric conditions along the flight path from the warm and moist Caribbean to the much drier and cooler condition inland provides a unique opportunity to address

the sensitivity of ARO measurements to the tropospheric temperature and moisture changes. Moreover, during the ferry flight, the ARO signals were simultaneously recorded from two high-gain antennas mounted on both sides of the aircraft fuselage, and one relatively low-gain antenna on the top of the aircraft (Haase et al., 2014, Murphy et al., 2015; Wang et al., 2016). Continuous measurements were made from all 3 antennas during the entire flight. Such a unique dataset offers a great opportunity to evaluate the precision of ARO measurements as well as the impact of signal tracking (e.g., antenna gain,

SNR) and different retrieval algorithms on the ARO retrievals.

The paper is structured as follows: Section 2 describes the details of the ARO measurements collected during the ferry flight of the PREDICT campaign. In addition, the other independent datasets used to validate the ARO retrievals, including the reanalysis data, radiosonde and COSMIC RO soundings are also introduced. Section 3 presents the atmospheric conditions over the study area from reanalysis data. Section 4 evaluates the ARO sounding quality by directly comparing

with all three independent datasets. The ARO retrieval differences resulting from various ARO retrieval algorithms (GO, FSI and PM) are also presented. The precision of ARO measurements is also quantified through comparison between the ARO recording from the top and side-looking antennas. Finally, the summary and conclusions are presented in Section 5.

## 2    Data and Methodology

### 2.1    ARO measurement

The NSF/NCAR Gulfstream V (HIAPER) aircraft, with the GNSS Instrument System for Multistatic and Occultation Sensing (GISMOS) onboard, was deployed in August–September 2010 for the Pre-Depression Investigation of Cloud

systems in the Tropics (PREDICT) field campaign (Montgomery et al., 2012; Haase et al., 2014; Murphy et al., 2015). A total of seven antennas were mounted on the exterior of aircraft including one on the top and two at each side of the fuselage for occultation measurements as well as two on the bottom of the fuselage for GPS reflection measurements (Garrison et al., 2007). During each flight, GISMOS continuously recorded GPS signals at 5 Hz from geodetic quality dual-frequency Trimble NetRS GPS receivers and simultaneously at 10 MHz using the GNSS Recording System (GRS). In addition to the





setting occultation measurements, the GRS doubles the number of ARO soundings by enabling open-loop (OL) tracking technique, which is capable of recording the rising occultation, and allows high-quality signal tracking in the moist lower troposphere (Wang et al., 2016). An Applanix POS/AV™ inertial navigation system was used to achieve velocity precision better than 5mm/s (Muradyan, 2012; Muradyan et al., 2010) as required for airborne retrieval accuracy to be better than 0.5

% in refractivity (Xie et al., 2008).

To evaluate the ARO sounding quality, we focused on the ARO measurements from three antennas, including one wide-view avionics antenna mounted on the top of the fuselage (CH1) that was intended to provide data for precise navigation and high elevation angle satellite data for clock corrections, and two high-gain antennas, with the gain patterns focused on the horizon for extra sensitivity in tracking low SNR occulting satellites, mounted on both the port (right; CH2) and starboard

(left; CH3) side of the fuselage. The GRS samples the wide-band GPS signals at 10 MHz on both L1 and L2 frequencies on these three channels. The wide view angle of the top antenna was found to provide sufficiently high SNR recordings of all ARO occulting satellites that were simultaneously recorded by either the port or starboard antenna. Such simultaneous recording between the top (CH1) and side antenna (CH2 or CH3) offers an excellent opportunity to access the precision of the ARO soundings.

Flight level in situ measurements of temperature were made at 50 Hz with a fast response, all weather, de-iced avionics sensor (Rosemount Model 102AL TAT) with 0.5°C accuracy (http://www.hiaper.ucar.edu/handbook/index.html). In situ pressure was measured at flight level with 0.1 hPa accuracy. Humidity measurements were made with a vertical cavity surface emitting laser hygrometer and a Buck research model 1011c hygrometer. However, because of the inconsistency of recording accuracy of the humidity data (Murphy et al., 2015), the in situ measurements were not used in this paper. The

sensitivity of in situ refractivity on ARO soundings are only limited to the ARO retrieval close to the aircraft altitude and has been well understood from previous studies (e.g., Xie et al., 2008, Murphy et al., 2005). Therefore, in this paper, the ECMWF reanalysis data interpolated to the ARO receiver position was used to compute the refractivity at the receiver.

### 2.1.1   Aircraft flight path and ARO soundings

The HIAPER aircraft was deployed for 26 research flights to studying eight storm systems during the PREDICT field campaign. The aircraft typically flown in a lawnmower or square spiral patterns to allow regular sampling of the development region of the targeted storm system. The quality of ARO soundings collected during the research flights has been analyzed in detail (Murphy et al., 2015). However, the frequent direction and altitude changes during the research flight

complicated the ARO signal tracking, and led to degraded ARO sounding measurements especially during turns (Murphy et al., 2015; Wang et al., 2016).

After completion of the PREDICT research flights, ARO measurements were then continuously recorded during the return ferry flight from the field station at St. Croix in the US Virgin Islands to the UCAR facility at Broomfield, Colorado on October 2, 2010. During the ferry flight, the aircraft cruised at a steady altitude of approximately 13 km above mean-sea-





level (MSL) along a nearly straight flight path, which provided an ideal recording geometry for ARO measurements. Figure 1 shows the aircraft flight path and the tangent points (e.g., local sampling positions) of each recorded occultation. Note that each GPS satellite is named for its unique Pseudo Random Noise (PRN) number. Each occultation event is labelled by a pair of PRNs for the occulting GPS satellite and the high-elevation GPS satellite used to correct for the receiver clock errors. The

ARO tangent point locations were estimated using the geometric-optics ray-tracing method (e.g., Xie et al., 2008) assuming a 1-dimensionally varying atmosphere represented by the Climate Impact on Regional Air Quality (CIRA+Q) refractivity climatological model (Kirchengast et al., 1999). It is worth noting that the ARO measurements sample both the moist conditions over the ocean during the early stage of the flight (15:00–16:40Z) and the relatively dry conditions over land during the final stage of the flight (e.g., 17:00–19:00Z). In the middle stage of the flight, the ARO senses the coastal region

over both land and ocean. The tangent points of three nearby COSMIC RO soundings within 6 h and 300 km of the ARO measurements are shown, labelled with the occulting GPS satellite number (e.g., G23, G29 and G05) along with the location of a radiosonde station in Lamont, Oklahoma used in profile comparisons.

[Figure 1]

Altogether, a total of 17 ARO events were recorded during the ~4-hour ferry flight. The wide-view angle top antenna (CH1) recorded all 17 ARO events, whereas the port (CH2) and starboard (CH3) high-gain antennas recorded 6 and 8 events, respectively. Table 1 lists all ARO events recorded during the ferry flight. Here the occultation period begins from tracking the occulting satellite at 5° positive elevation angle above the local horizon until the tangent point descends close to

the surface for a setting occultation, and vice versa for a rising occultation. Therefore the beginning time of each ARO event is defined as when the TP is either at positive elevation of 5° for a setting occultation or near the surface for a rising occultation. Note that the tangent point at zero elevation angle (local horizon) will be at the aircraft altitude. During a setting ARO event, the TPs will gradually descend and move away from the aircraft until the TP touches the surface (Xie et al., 2008), and vice versa for a rising occultation. The TP drift distance measures the distance between the location of the TP at

the aircraft altitude (zero elevation angle) and the TP near the surface at MSL. The TP location (latitude and longitude) for each ARO event when the TP is near the surface, is also shown. With an aircraft cruising at around 13 km, the average ARO sounding takes about 29 minutes with the shortest event observed during this campaign being about ~20 min (prn07-28) and the longest event being about ~44 min (prn15-28). Generally, the tangent point drifts away from the aircraft, but will vary in extent with variations in the relative positions and velocities of the occulting GPS satellite and the aircraft. The TP drift

distance varies from 181 km (prn08-19) to 589 km (prn06-11) with an average of about 375 km. Interestingly the four longest ARO events (i.e., over 33 min) actually have relatively short TP drift distance of less than 300 km.

[Table 1]



### 2.1.2 ARO retrievals methods

Because ARO faces the same problems with atmospheric multipath, similar radio-holographic retrieval methods have been implemented for ARO measurements to improve on the geometric-optics (GO) retrieval of bending angle (Xie et al., 2008, Haase et al., 2014, Murphy et al., 2015). These are the full-spectrum-inversion (FSI, Adhikari et al., 2016) and the phase-matching method (PM, Wang et al., 2017. A brief description of the three retrieval methods is presented below.

In the GO method, the GPS signals are considered as rays with unique impact parameters that describe the geometry of the raypath. With the assumption of a spherically symmetric atmosphere, the bending angle as a function of impact parameter can be uniquely determined from precise measurements of ARO receiver and GPS satellite positions along with the excess Doppler shift of the GPS signal. After removing the Doppler resulting from the movement of GPS transmitter and receiver, the excess Doppler due to the atmosphere can be derived and used for retrieving the bending angle and refractivity profiles (e.g., Lesne et al., 2002; Xie et al., 2008). In regions of highly variable refractivity gradients, which often occurs in moist lower troposphere, multiple rays with different geometry can arrive at the receiver at the same time and constructively and destructively interfere, which violates this assumption of a single ray. In that case, the GO method suffers limited vertical resolution and significant refractivity bias. To correctly distinguish these multiple signals, RH retrieval methods are needed.

The FSI method proposed by Jensen et al (2003) recognizes the RO signal recording as a summation of radio waves of different frequencies and accounts for their interference. Each wave with a unique frequency corresponds to one single ray path in the GO approach. Multiple frequencies present in the signal at a given time can be unambiguously identified by taking the Fourier transform of the RO signal and using the method of stationary phase. This FSI method has been successfully implemented in spaceborne RO retrievals and has significantly improved both the bending and refractivity retrievals as compared to the GO method. The FSI method was adapted for airborne RO simulation and described in Adhikari et al. (2016). FSI requires a perfectly circular trajectory for both transmitter and receiver. Therefore, a geometric correction to the phase is needed to account for the transformation from the real, non-spherical trajectories to circular trajectories referenced to a local center of curvature. The Fourier transform is then applied separately to both the negative and positive elevation angle segments of the ARO measurements to retrieve the ARO bending angle profile for each segment.

The phase-matching (PM) method (Jensen et al., 2004) is another RH method that also utilizes the method of stationary phase (MSP) to calculate the bending angle profile. Instead of frequency, the PM method uses impact parameter to identify individual subsignals. The impact parameter is forward modelled during occultation period considering the arbitrary receiver and GPS orbit geometry without the need for the correction required by the FSI method. The PM method was first adapted for ARO by Wang et al. (2017).

Once the ARO bending angle is retrieved from one of the three retrieval methods, a modified inverse Abel transformation can then be applied to retrieve the ARO refractivity profile (Xie et al., 2008). There is a singularity in the



ARO retrievals near zero elevation angle (close to aircraft height) where small errors in ray tangent angle can result in large bending angle errors near the receiver altitude, which can be constrained by the in situ refractivity measurement (Xie et al., 2008; Adhikari et al., 2016). These errors could propagate downwards and introduce refractivity bias in the  retrieved refractivity profiles. To mitigate this refractivity bias, the retrieved bending angle profiles at the top 1.5 km below the

receiver altitude are replaced by the simulated bending angle profiles obtained from collocated ECMWF reanalysis refractivity profile. As a result, the independent refractivity retrievals from ARO measurements extends from surface up to about 11.5 km, which is ~1.5 km below the receiver altitude.

In this paper, the main focus will be on the first results from the analysis of the ARO retrievals using FSI. Comparison among all three retrieval methods will also be presented in Sect. 4.4.

## 2.2    ECMWF reanalysis and radiosonde data

To evaluate the ARO sounding quality, the high-resolution, 6-hourly European Centre for Medium-Range Weather

Forecast (ECMWF) Reanalysis – Interim (ERA-I) data were used. The spatial resolution of the reanalysis data is approximately 80 km (T255 spectral) with uniform grid (0.75° latitude x 0.75° longitude) on 60 vertical levels from the surface up to 0.1 hPa (Dee et al., 2011). The vertical grid levels are unevenly distributed with more levels at lower altitudes. About half of the model levels (28) are below 10 km, of which 21 levels are below 5 km, and 14 levels are below 2 km. The vertical grid interval increases at higher altitudes, from less than 200 m below 1 km to ~500 m near 5 km.

Given the ERA-I temperature, pressure and mixing ratio data, the corresponding refractivity ($N$) can be calculated (Smith and Weintraub, 1953; Healy, 2011). The simulated bending angle profile can be calculated through the Abel integral of the refractivity profile (Xie et al., 2008). Both the ERA-I refractivity and bending angle profiles can then directly compared with the near-coincident ARO soundings. Due to the long tangent point drift (~375 km) (Fig. 1), the ARO sounding senses the atmosphere over a large area, which could cover multiple ERA-I grid cells. Therefore, the ERA-I file

within 3-hours of the ARO sounding is first identified, then the refractivity at each ARO TP location is derived through 3-dimentional bilinear interpolation of the surrounding eight ERA-I grid values.  The comparison is expected to produce an estimate of the combined error of the ARO measurement and the effect of horizontal model variations integrated over the entire ray path.

In addition, the Department of Energy (DOE) Atmospheric Radiation Measurement (ARM) program Central Facility

(Latitude: 36.62°N, Longitude: 97.48°W, Elevation: 317 m), located near Lamont, in north-central Oklahoma, provides data for validation purposes. The core instrumentation at the Southern Great Plains (SGP) site provides radiosonde soundings four times daily, and continuous measurement of surface temperature, pressure and precipitable water vapor from a microwave radiometer, MWR. For several ARO soundings, two close-by radiosonde soundings were identified, with one at 17:28Z and the other at 23:28Z on October 2, 2010.





### 3    Atmospheric conditions over the study area

As shown in Table 1, ARO soundings have relatively long duration (~30 min) and large TP drift (~375 km). The horizontal variation of the atmosphere needs to be assessed to better understand the ARO measurements. The ARO

soundings can be separated into two categories, ones over land (6 soundings) and the others over ocean, including those over coastal regions (11 soundings), based on their tangent point locations. The ARO soundings in the earlier stage of the flight were taken mostly over the ocean/coastal region (1400–1640Z), and the soundings at the later stage are mostly over land (1700–1900 Z).

Figure 2 shows the ERA-I temperature, moisture and the refractivity field at two pressure levels (850 hPa and 500 hPa),

at 18Z on October 2, 2010. The synoptic pattern was dominated by an upper-level trough stretching from central eastern Canada to the Midwestern United States as seen in the 500 hPa height contours (Fig. 2e). A cold front was located over Oklahoma and orienting northwest across the south eastern US all the way to West Virginia. On the contrary, high values of moisture extended from low latitudes up to ~25°N over the Caribbean.  As a result, very cool and dry conditions were found at the SGP site near the end of the ferry flight, whereas much warmer and more moist conditions were found over the

Caribbean (Fig. 2a,b). The slow movement of the cold front into the southern US led to sharp changes in the atmospheric conditions over this region during the 4 h ferry flight. The large spatial variations in temperature and moisture led to significant inhomogeneity in the refractivity field. Two regions (Fig. 2c,f) were selected to contrast the atmosphere over the warm and moist Caribbean [67–78° W, 18–27° N] with that of the cool and dry land near the Southern Great Plains of the US  [89–99° W, 32–40° N]. It is worth noting that the two selected regions do not include the Gulf Coast of Mexico and

Florida, which exhibit warm and moist lower troposphere similar to the Caribbean, but rather dry upper troposphere similar to inland region (Fig. 2).

[Figure 2]

The vertical profiles of the mean moisture and refractivity from ERA-I along with their anomalies and root-mean-square (RMS) difference, over the two selected regions are shown in Figure 3. Over land (89–99° W, 32°–40° N), rather dry atmospheric conditions (mean mixing ratio < 6 g/kg) with low near-surface refractivity (< 300 N-unit) are observed in the northwestern domain (Fig. 3a,c). The planetary boundary layer (PBL) was moist below ~3 km with mixing ratios of less than 1.5 g/kg above this level and a maximum reaching ~6 g/kg near the surface. Most variation in moisture is seen below 3 km

with a maximum around 2 km, where the RMS difference is close to 2 g/kg in mixing ratio, resulting in a large variation in refractivity of up to ~15 % (Fig. 3a,c). Above 10 km, water vapor content is low (< 0.05 g/kg), and the relatively large variation of refractivity (RMS: ~1 %) indicates the variability in the upper troposphere over land. It should be noted that the sharp decreases in moisture around 3 km and 1 km leads to large refractivity gradients at both altitudes, which can introduce multipath propagation and result in larger differences between the GO and RH refractivity retrievals.



Over the ocean domain (67–78° W, 18–27° N), a much more moist atmosphere, extending up to the upper-troposphere, is observed than over the land. The moisture exponentially decreases with altitude from a maximum of ~17 g/kg near the warm ocean surface (with a high surface refractivity of ~380 N-unit). The troposphere near 7 km altitude remains moist with mixing ratio ~1.5 g/kg (Fig. 3b). The highest moisture variability occurs near 5 km, where the maximum RMS difference

reaches ~1.5 g/kg in mixing ratio, and ~7 % in refractivity. The refractivity above 10 km shows much less variation over the ocean with a RMS difference (~0.5 %) only half that found over land (~1 %), where the upper troposphere variability occurs significantly higher in the profile.

[Figure 3]

## 4    Evaluation of the ARO retrievals

To evaluate the quality of the ARO measurements, the near-coincident ERA-I reanalysis, radiosonde and COSMIC RO soundings were directly compared to the ARO soundings. A quantitative assessment of the ARO retrievals is made through

the inter-comparison of three different retrieval methods. Moreover, the precision of ARO measurements is evaluated by comparing the same ARO event from the top and side-looking antennas recorded on two independent channels during the ferry flight.

### 4.1    ARO retrievals with near-coincident ERA-I profiles

All 17 ARO measurements recorded from the top antenna (CH 1) were processed with the FSI method to retrieve the ARO bending angle profiles, which can be further used for deriving the refractivity profiles. A typical ARO event over Alabama, labelled prn15-28 for occulting satellite 15 corrected by subtracting the residuals from satellite 28, along with the near-coincident ERA-I profile are presented in Figure 4. Note the ERA-I bending angle profile is simulated based on the

forward Abel integration of the refractivity (e.g., Xie et al., 2008). The bending angle is a function of impact parameter, which is the product of the refractivity index and the radial distance of the tangent point from the center of local curvature. For illustration purposes, the impact height is used, which is simply the difference between the impact parameter and the radius of curvature. Because impact height depends on refractivity, it is generally a value of about 2 km at the surface in the tropics.

The ERA-I profile shows a weak temperature inversion with a large moisture gradient near 1 km, which leads to a large refractivity gradient and a sharp increase in bending angle around an impact height of 3 km. Both the ARO bending angle and refractivity profiles are highly consistent with the ERA-I profiles with the mean refractivity difference of about ~0.2 % (RMS 1%)  overall and less than 0.1 % (RMS 0.5 %) above 3 km. The ARO sounding retrieved from FSI also captures the



PBL height well at about 2.5 km. Larger differences are seen in refractivity below that and in the bending angle below ~4 km impact height.

The prn15-28 occultation is a rising case, where the ARO receiver tracks the GPS occultation signal from near the surface to the upper atmosphere. For example, the signal to noise ratio (SNR) around UTC 17.1 h is close to the background

noise, when the tangent point is near the surface. Note a sharp drop of ARO SNR at 17.17 h to the background noise, and the strong signal re-emerging at ~17.12 h in Fig. 4c The large SNR variation is a strong indication of signal interference due to multipath resulting from the sharp refractivity gradient seen in the ERA-I profiles near 1 km.

[Figure 4]

After comparing each ARO refractivity profile from channel 1 with its near-coincident ERA-I profile, the fractional refractivity differences are shown in Figure 5. Overall, the ARO profiles are highly consistent with the ERA-I above ~5 km with near zero bias of –0.13 % (RMS 0.21 %) in the middle and upper troposphere. In the lower troposphere, however, the ARO refractivity shows a negative bias of about –1.5 % (RMS 1.7 %) below 5 km with a maximum of –3 % near the

surface. As large differences in atmospheric conditions are seen between land and ocean (Fig. 3), we further separate the ARO soundings into two categories with one group of ARO soundings over land (i.e., with all TPs over land) and the others over ocean (i.e., with partial or all TPs over ocean) as shown in Table 1. The ARO profiles over ocean show a negative refractivity bias below 5 km, where large moisture variations begin in the ERA-I profiles over the ocean (Fig. 3b,d). Similarly, the ARO profiles over land show the negative refractivity bias below ~3.5 km with a maximum bias near 2 km,

where the maximum moisture and refractivity variations are observed in ERA-I profiles over land (Fig. 3a,c). Overall, the negative ARO refractivity biases in the lower troposphere seem to be related to the moisture variations. Wang et al. (2016) showed that when refractivity is higher than the climatological value used in the Doppler model for the open-loop tracking, low SNR could potentially lead to an unwrapping error in the carrier phase measurements that would produce a preferentially negatively biased refractivity.

[Figure 5]

### 4.2    ARO retrievals with near-coincident radiosonde measurements

Near the end of the ferry flight, there are two ARO profiles (prn09-28 and prn07-28) near the SGP site at Lamont, Oklahoma (Fig. 1), where radiosonde profiles were launched at 17:28Z and 23:28Z on October 2, 2010 (Fig. 6). The vertical profiles of temperature, relative humidity and the derived refractivity for the two soundings are shown in Fig. 6a,b. The radiosonde around at 17:28Z, or 11:28 PM local time, shows a complicated multiple layer structure. Three distinct layers marked by high relative humidity gradients and weak inversions are seen at around 1.3 km, 1.7 and 2.7 km, where large



negative refractivity gradients are also present. On the other hand, the radiosonde in the late afternoon (at 23:28Z, or 5:28 PM) shows a well-defined single layer PBL with a sharp inversion and large relative humidity and refractivity gradient at ~1.9 km. Note that the sharp refractivity gradient exceeds the critical refraction of −157 N-unit/km and leads to a ducting layer across the PBL inversion layer.

The time series of the surface temperature and relative humidity as well as the precipitable water vapor (PWV) from the microwave radiometer shows the high pressure system was moving into the area. The cold front caused significant change around 18Z and a strong subsidence and drying afterward, which creating a stronger boundary layer inversion at the radiosonde station in late afternoon (Fig. 6c,d). Near local noon (18:00Z), relatively stable surface temperature but a rapid decrease in PWV are observed. During the one hour time span from 17:30Z to 18:30Z, the PWV decreased by 38 % from 1.6

to 1.0 cm (a decrease of 0.6 cm, or ~ 6 kg/m$^2$). The PWV further decreased to ~0.7 cm at 19:00Z and remained rather constant into the late afternoon (23:00Z, or 5PM local time). Most of the water vapour is found in the boundary layer as seen in Fig. 6a,b. The significant change in the PWV near the radiosonde launch time at 17:28Z but rather small variation at 23:28Z, implies larger temporal variation of the lower tropospheric (or PBL) structure near local noon time (17:28Z) as compared to the late afternoon (23:28Z).

[Figure 6]

The bending angle profiles of the two ARO measurements from prn07-28 (at 18:57Z, Fig. 7a) and prn09-28 (at 18:19Z, Fig. 7b) are presented along with the simulated bending angle profiles from the collocated radiosonde and ERA-I profiles at

17:28Z and 23:28Z, respectively. The refractivity profiles and the difference between the ARO and the radiosonde/ERA-I profiles are also shown in Fig. 7c,d, respectively. The two radiosonde soundings are almost identical in refractivity above ~4 km (0.7% RMS) but show large differences (up to ~15 % near 2.5 km) inside the boundary layer (Fig. 7d) due to the strong temporal variation of atmospheric conditions resulting from the synoptic forcing and the local diurnal surface heating changes (Fig. 2 and Fig. 6).

The radiosonde at noon (17:28Z) shows three jumps in bending angle at ~3, 4 and 4.5 km impact height and a small increase in bending at ~6 km. On the other hand, the late afternoon sounding (at 23:28Z, or 6:28 PM) shows only one large jump in bending angle at ~3.5 km impact height (Fig. 6d). It is important to note that even though the two ARO soundings are collected around the same time, the ARO prn09-28 sampled the PBL in northern Oklahoma where the cold air mass associated with the high pressure system was already dominant, whereas the ARO prn07-28 sampled the PBL in southern

Oklahoma, when the cold front had just moved in and caused dramatic changes as seen in the in situ measurements at Lamont, Oklahoma (Fig. 6).

The ARO profile (prn07-28) is about 1.5 hours later and 290 km south of the radiosonde (RDS1728Z). It also detects distinctly sharp bending angle near 3 km, 4 km, 4.5 km and 6 km impact height (Fig. 7a). These correspond to the bending angles jumps seen in the radiosonde (RDS1728Z) but with slightly underestimated bending beneath each layer. Despite



capturing the height of the individual layers, and showing agreement above 8km (~0.8 % RMS in refractivity), the ARO and the radiosonde refractivity differences are significant at lower levels, showing RMS difference ~2 % in the height range 4-8 km, and a maximum difference up to −9 % at ~2.5 km. In addition, a slightly better agreement between the ARO and the ERA-I profiles is seen.

The ARO profile (prn09-28) is a similar distance, about 250 km north of the radiosonde site, but is about 5 hours earlier than the radiosonde (RDS2328Z) observation. However, remarkable consistency between the two in both bending angle and refractivity are shown above 2 km (less than 0.5 % RMS in refractivity), with a large negative bias (up to −10 %) in refractivity below ~2 km. This signature in the bending angle profile is typical in the presence of a ducting layer across a sharp inversion layer (Fig. 6), which could introduce large negative refractivity biases in standard Abel retrieval due to the
non-unique inversion problem (Sokolovskiy 2003; Xie et al., 2006). The ARO sounding shows one large bending angle jump near 3.5 km impact height, corresponding to the sharp refractivity gradient near 2 km that is also observed by the radiosonde. Beneath this height, the ARO bending angle is smaller than the radiosonde bending, which results in a negative refractivity bias below 2 km. Moreover, the ARO profile agrees extremely well with the collocated ERA-I profile, including a much smaller difference inside PBL below 2 km (Fig. 7d). Both the ARO and the collocated ERA-I profile show a smaller
refractivity gradient without a ducting layer as observed in the radiosonde profile (RDS2328Z) near 2 km. The better agreement between ERA-I and the ARO measurement implies the likely presence of the horizontal inhomogeneity inside the PBL over the region, where the fine vertical structure observed from the in situ radiosonde might not be representative of a large domain (e.g., 100-200 km horizontally), and could be smoothed out in the ARO observation.

[Figure 7]

### 4.3    ARO retrievals compared to nearby COSMIC RO soundings

    During the ferry flight, three COSMIC RO soundings occurred within 300 km of the ARO soundings and within a 6-
hour time window (Fig. 1). Two ARO soundings near 18Z over land (prn 24-28 and prn09-28, see Table 1) are paired with one COSMIC RO sounding (G05) at ~14Z. Another two ARO soundings near 15Z over the ocean (prn01-19 and prn 08-19) are paired with two COSMIC soundings (G29 and G23) at 12Z and 17Z, respectively. For direct comparison of COSMIC RO bending angle to the ARO retrieval, the COSMIC bending angle is calculated from the observed COSMIC refractivity profile using the same forward Abel integral (Xie et al., 2008) used for ERA-I and radiosonde profiles (Fig. 8).
Figure 8(a, c) show the bending angle and refractivity comparison between the two ARO profiles and COSMIC profile over land (within ~4 h in time and 200 km in space). The COSMIC sounding (G05) is located north of the ARO measurements. Both ARO soundings (prn24-28 & prn09-28) show a sharp bending angle gradient at ~3 km impact height. In contrast, the COSMIC sounding (G05) detects two sharp bending angle layers with one at slightly lower than 3 km and another one at ~4 km. The two ARO refractivity profiles show a better agreement with the COSMIC profile (G05) above ~3



km with RMS refractivity difference of ~1.8 % for prn09-28, and ~2 % for prn24-28. At the top of the ARO profile at ~13 km where the ARO refractivity is constrained by the ERA-I reanalysis, the large difference (~4 %) between the ARO and COSMIC (Fig. 4c) indicates that the much cooler upper troposphere observed in the COSMIC sounding location in the north of the ARO sounding could be caused by the synoptic condition difference. The mis-alignment of the bending angle variations in the boundary layer  between the ARO and the COSMIC profiles leads to a large negative refractivity difference of −8 % near the surface for both ARO profiles.

Figure 8(b, d) show the two ARO soundings (prn01-19 and prn 08-19) at ~15Z (10 AM local time) and the nearby COSMIC profile (G29) over the Caribbean (within ~1 h in time and ~300 km in space). The other COSMIC profile (G23) occurred about 2.5 hours earlier but over 500 km away from both ARO profile. The two COSMIC soundings agree within less than 1.5 % RMS refractivity difference above 3 km. However, large difference in refractivity are seen at lower levels with a maximum of −8 % near 2.5 km. Overall, the two ARO refractivity profiles show a better agreement with the COSMIC profile (G29) above ~3 km with RMS refractivity difference of about 1.6 % for prn01-19, and 1.5 % for prn08-19 (Fig. 8d). Note the two ARO profiles were collected ~ 2-hour later than G29 but ~2-hour earlier than G23. A sharp increase in bending angle is observed in COSMIC (G23) at  ~3.5 km impact height (or around 2 km in height) but not in G29. The large bending angle and sharp refractivity gradients observed in both the ARO and COSMIC soundings in the lower troposphere can probably be attributed to the fine vertical structure in boundary layer moisture variation. The larger differences in bending angle structure in the mid-troposphere seen in prn01-19 and prn 08-19 over the ocean (Fig. 8d), relative to prn09-28 over land (Fig. 8c), confirms the higher variability in moisture at the mid-troposphere over ocean.

Overall, the comparable RMS refractivity differences between the ARO and nearby COSMIC profiles (~1.5-2 %) with that of the two nearby COSMIC profiles (~1.5 % between G23 and G29) implies ARO could achieve comparable quality sounding as the spaceborne RO. The sensitivity of the ARO sounding to the local moisture and temperature variations along with the high spatial density of soundings offers a great complement to the spaceborne RO soundings for mesoscale and transient weather event studies.

[Figure 8]

### 4.4    Comparison of ARO profiles derived from different retrieval methods

As described in Section 2.1.2, there are three major ARO retrieval methods, including the geometric-optics and two radio-holographic methods that were used to retrieve the bending angle profiles. The ARO measurements recorded from channel 1 are separated once again into land and ocean categories, based on the tangent point locations. Figure 9 shows the mean difference between the ARO profiles from the three methods and the near-coincident ERA-I profiles using ERA-I as the reference. The difference for both the ARO bending angle (Fig. 9a,b) and refractivity (Fig. 9c,d) retrievals over land and



ocean are shown separately. Furthermore, the fractional refractivity differences between the GO/PM retrievals and FSI are shown in Fig. 9e,f.

The ARO bending angle profiles from all three retrieval methods display small difference from the near-coincident ERA-I profiles above ~6.5 km over land (Fig. 9a) and above ~8 km over ocean (Fig. 9b) in impact height. Correspondingly,
the small RMS fractional refractivity difference is about 0.3 % above ~5 km over both land and ocean (Fig. 9c,d). Below these levels, all three retrieval methods show negative biases increasing at lower altitude. The GO retrieval deviates significantly from the FSI and PM retrievals and shows larger negative biases in bending and refractivity due to the apparent multipath problem resulting from the increasing moisture in the lower troposphere (e.g., Murphy et al., 2014). The higher transition altitude for large GO retrieval bias indicates the multipath problem is worse for GO over ocean (~7 km) than land
(~5.5 km) as may be expected from the height dependence of the moisture variability in Fig. 3. All three ARO refractivity (and bending) retrievals reach the maximum biases at around 2.5 km over land and 4 km over ocean, where the maximum moisture variations are observed in the lower troposphere, respectively (Fig. 3). The RMS refractivity difference between PM and FSI is less than 0.1 % above 5 km and increases slightly at lower levels over land. The negative refractivity bias below 5 km is slightly greater for PM retrievals over ocean compared to FSI. However, given the variability and the small
number of profiles over ocean analyzed in Figure 9f, this may not be significant.

The remaining negative bending angle and refractivity biases in the moist lower troposphere in FSI and PM retrieval (Fig. 9c,d) may be due to low SNR in the lowest levels of the atmosphere. The effects of SNR can be examined by investigating the differences among antennas that are described in the next section.

[Figure 9]

### 4.5      ARO antenna evaluation and measurement precision

Out of the 17 ARO events recorded from the top antenna (Table 1), there are 12 ARO events that are simultaneously
recorded from one of the two side-looking antennas. The others occurred with a viewing azimuth directly fore or aft of the aircraft. Four were recorded from the port (CH2) and eight from starboard (CH3) antenna (Table 1). The redundant measurements provide a unique opportunity to estimate the precision of the ARO measurements and evaluate the key factors affecting the GPS occultation signal tracking. Note the isotropic top antenna had much better recordings of the high-elevation GPS satellites than the side-looking antenna. For each ARO event, the same high-elevation GPS satellite tracked
by the top antenna was used for clock calibration of the occultation measurements from both the top and side-looking antennas. The same processes were applied to the individual channels from side and top antennas for the open loop tracking, filtering, FSI retrieval and Abel inversion to derive the bending angle and refractivity profiles.

Figure 10 shows the difference between the ARO refractivity retrievals from the side and the top antennas. The individual refractivity difference profiles for each ARO event and the mean difference are shown. Note there is one obvious



outlier (prn16-19), which shows a large negative refractivity difference below 3 km reaching −10 % near the surface. Without considering the outlier case, the RMS refractivity difference is less than ~0.1 % above 3 km and 0.2 % overall from the surface up to ~13 km. There is a small positive bias from 0 to 4 km, with a maximum of ~0.5 % near 3 km. The small retrieval difference between the ARO measurements from the top and the side-looking antennas indicates high precision in

the raw ARO measurements can be achieved, corresponding to an RMS better than 0.2 % in refractivity (or ~0.4 K).

[Figure 10]

The refractivity retrieval differences between the top and side antennas for each ARO event (Fig. 10), are likely caused

by differences in the SNR of the two ARO signals, which results in noise variations in the phase extracted in the open-loop tracking procedure (Wang et al., 2016). Although the side-looking antennas had higher gain in the view direction perpendicular to the aircraft, the line of sight to the GPS satellite was rarely in that direction and often appeared in part of the antenna pattern with lower gain. The antenna directivity limits has lower gain in the forward and aft directions. However, the wide-view top antenna generally maintains high SNR until the lowest elevation angle measurements when the tangent

point descends into the lower troposphere. It permits observations of GPS satellites at the full 360-degree range of azimuth angles.

Figure 11(a, c) shows the outlier case (prn16-19) from Fig. 10 that displays large negative refractivity difference between the side (starboard) antenna (CH3) and the top antenna (CH1). The SNR measured from both antennas are shown in Fig. 11a. In comparison to the top antenna, the side-looking antenna shows generally lower SNR with large variation during

the setting occultation. It started tracking the prn16 at ~140° azimuth angle relative to the flight heading direction at 15.75 hrs (UTC), i.e., about 40° from the back of the aircraft. The azimuth angle further increases to over ~145° near 15.9 hrs, which led to a significant drop of the SNR close to the noise background. The SNR thereafter recovered slightly but dropped again to near the noise background (~20 v/v) at around 16.16Z with azimuth angle over 150°, whereas the top antenna maintained relative high SNR above ~80 (v/v). Such low SNR can result in incorrectly resolved phase unwrapping and can

introduce a bias, usually negative, in the reconstructed signal phase (Fig. 11c), which further leads to a negative bias in the ARO bending and refractivity retrievals (Wang et al. 2016).

On the other hand, a typical normal case (prn01-19) is also shown (Fig. 11b,d). With the occulting GPS satellite in the boresight of the side-looking antenna (e.g., azimuth angle of ~138-146° relative to flight heading direction), the SNR from the side-looking antenna shows higher SNR than the top antenna. The excess phase difference between the two antennas

remains very small during the ARO event and only has some small differences in the lowest troposphere (Fig. 11d), which does not introduce significant differences in the bending angle and refractivity retrievals.

[Figure 11]



## 5    Summary and Conclusions

The airborne radio occultation technique can offer dense RO soundings over targeted regions during mesoscale and
transient weather events, with comparable data quality to the spaceborne RO soundings in the mid to upper troposphere. In
this study, the airborne radio occultation measurements from the 4 h long (~3600 km) ferry flight from the Virgin Islands to
Colorado at the end of the PREDICT field campaign were analysed, and the quality of the ARO retrievals were evaluated.
During the ferry flight, the aircraft was cruising at a steady altitude of approximately 13 km along a nearly straight flight
path, which provided the ideal recording geometry for ARO measurements from side-looking antennas. The ARO sampled
the warm and moist Caribbean environment to the cool and dry continental environment near the Southern Great Plains of
the US. A total of 17 ARO soundings were recorded by the top antenna, among which 12 ARO soundings were
simultaneously recorded by the side-looking antennas. The ARO soundings take an average of 29 minutes to sense from 5°
above the aircraft local horizon down to the surface with an average of TP drift of 375 km. The geometric-optics (GO)
method, the full-spectrum-inversion (FSI), and the phase-matching (PM) methods are used to retrieve the ARO bending
angle profiles, which are then used to derive the refractivity though the modified Abel inversion (Xie et al., 2008). The
quality of ARO sounding profiles is assessed in detail by comparison with the near-coincident independent datasets,
including the ERA-I reanalysis, radiosondes, and COSMIC soundings.

Comparison of ARO FSI refractivity retrievals with the ERA-I profiles interpolated to the tangent point locations
shows near zero refractivity bias of –0.13 % (RMS < 0.21 %) in the middle and upper troposphere from 5 km up to the
aircraft altitude at ~13 km. In the lower troposphere, however, the ARO refractivity shows a negative bias of about –1.5 %
(RMS 1.7 %) below 5 km with a maximum bias near 4 km over ocean and near 2 km over land (Fig. 5), corresponding to the
altitude of maximum moisture gradients for ocean and land (Fig. 3), respectively.

Two high-resolution radiosonde soundings from Lamont, Oklahoma, were compared to the nearby ARO
measurements. The two radiosonde soundings are 6 hours apart and show almost identical refractivity above ~4 km but
larger differences (up to ~15 % in refractivity) inside the boundary layer due to the strong PBL variations over land (Fig. 7).
Although ARO measurements are 290 km from the radiosondes, both ARO profiles agree well with the radiosonde above 4
km and capture the heights of sharp layers in the PBL observed by the radiosondes  (Fig. 7). The underestimation of the
ARO bending angle in the PBL leads to a negative bias in refractivity compared to the radiosonde. However, the smaller
difference between ARO and ERA-I profiles indicate that ARO succeeds in representing the refractivity at this larger scale
domain. The ARO soundings do not compare as well with the closest COSMIC RO soundings from the limited samples. But
the comparable RMS refractivity differences between the ARO and nearby COSMIC profiles (~1.5-2 %) with that of the two
nearby COSMIC profiles (~1.5 %) implies ARO could achieve similar quality sounding as the spaceborne RO.





The ARO retrieval uncertainty due to the bending angle retrieval method is also analyzed. The ARO profiles from all three retrieval methods display small RMS refractivity difference with less than 0.3 % above ~5 km over both land and ocean, compared with the near-coincident ERA-I profiles (Fig. 9c,d). Below these levels, all three retrievals show negative biases which increase at lower altitude. The FSI and PM retrieval methods significantly reduce the negative refractivity bias

seen in the GO retrieval by addressing the multipath problem.

Analysis of the 12 ARO events that were simultaneously recorded from the top and side-looking antennas shows that highly consistent ARO measurements are achieved. The overall RMS refractivity difference is less than ~0.1 % above 3 km and ~0.2 % from the surface up to ~13 km. There is a small positive bias below 4 km with a maximum up to ~0.5 % near 3 km. The small retrieval difference between the ARO measurements from the top and the side-looking antennas indicates

high precision in the raw ARO measurements can be achieved, corresponding to an RMS difference better than 0.2 % in refractivity (or ~0.4 K). One outlier case shows the low SNR recorded from the side-looking antenna results in errors in the carrier phase unwrapping which introduces a negative bias in the reconstructed signal phase, and further leads to a negative bias in the ARO bending and refractivity retrievals (Wang et al. 2016).

In summary, the ARO measurements by GISMOS from this ferry flight demonstrate its capability of providing

relatively dense soundings for targeting synoptic to mesoscale weather systems. The radio-holographic retrieval methods significantly improve the ARO retrieval in the moist lower troposphere where frequent multipath occurs and otherwise causes large negative biases in the GO retrieval. The ARO soundings capture well the height of sharp refractivity gradients in the moist lower troposphere, especially inside the PBL. The remaining negative bias in ARO bending angle and refractivity in the moist lower troposphere is most likely a result of low SNR, that may be best addressed with improved

antenna design. The data reveal the presence of sharp moisture gradients and boundary layer ducting phenomena that warrant further attention.

**Acknowledgements**

Funding for this research was provided by NSF grant AGS-1262041 and NASA grant NNX15AQ17G. Jennifer S. Haase

was supported by NSF grant AGS-1015904 and NASA grants NNZ12AQ86G/NNX12AK30G. We also thank Dr. Michael Murphy from the University of California, San Diego for valuable suggestions to improve the manuscript. COSMIC GPS RO soundings were obtained freely from the University Corporation for Atmospheric Research (http://cdaac-www.cosmic.ucar.edu/cdaac/products.html). The in situ surface measurement and radiosonde data were provided by the Department of Energy (DOE) Atmospheric Radiation Measurement (ARM) program. ERA-Interim reanalysis profiles were

provided by the European Centre for Medium Range Forecasts (ECMWF).



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



**Table 1: List of all ARO events recorded during the ferry flight on October 2, 2010, with the ARO profiles over land shown in bold.**

| prn # | Beginning time (UTC hours) | Duration (min) | Channel | Rising/ Setting | Latitude (°N) | Longitude (°W) | TP drift (km) |
|---|---|---|---|---|---|---|---|
| prn28-19 | 14.68 | 33.6 | 1 | R | 19.76 | -72.06 | 263 |
| prn08-19 | 14.76 | 24.0 | 1, 3 | R | 23.39 | -71.47 | 181 |
| prn31-19 | 14.83 | 23.4 | 1 | S | 18.95 | -68.19 | 287 |
| prn01-19 | 14.94 | 36.6 | 1, 3 | S | 23.42 | -69.17 | 291 |
| prn16-19 | 15.72 | 35.4 | 1, 3 | S | 25.74 | -74.59 | 255 |
| prn17-19 | 15.31 | 27.6 | 1, 2 | R | 19.26 | -75.34 | 504 |
| prn06-11 | 16.12 | 32.4 | 1, 3 | S | 30.50 | -77.34 | 589 |
| prn23-11 | 16.52 | 25.8 | 1, 2 | S | 24.93 | -82.35 | 337 |
| prn03-07 | 16.73 | 30.0 | 1, 3 | S | 32.59 | -80.65 | 497 |
| **prn15-28** | 17.08 | 44.4 | 1 | R | 33.45 | -86.80 | 270 |
| prn13-07 | 17.24 | 27.0 | 1, 2 | S | 27.95 | -88.00 | 390 |
| prn04-28 | 17.46 | 29.4 | 1, 2 | R | 27.25 | -88.56 | 562 |
| **prn27-28** | 17.76 | 21.6 | 1 | R | 35.66 | -91.42 | 328 |
| **prn19-28** | 17.97 | 27.6 | 1, 3 | S | 37.08 | -89.16 | 441 |
| **prn24-28** | 18.10 | 30.0 | 1, 3 | S | 38.85 | -91.96 | 560 |
| **prn09-28** | 18.32 | 21.6 | 1, 3 | R | 37.97 | -95.61 | 260 |
| **prn07-28** | 18.96 | 19.8 | 1, 2 | S | 34.00 | -98.08 | 356 |





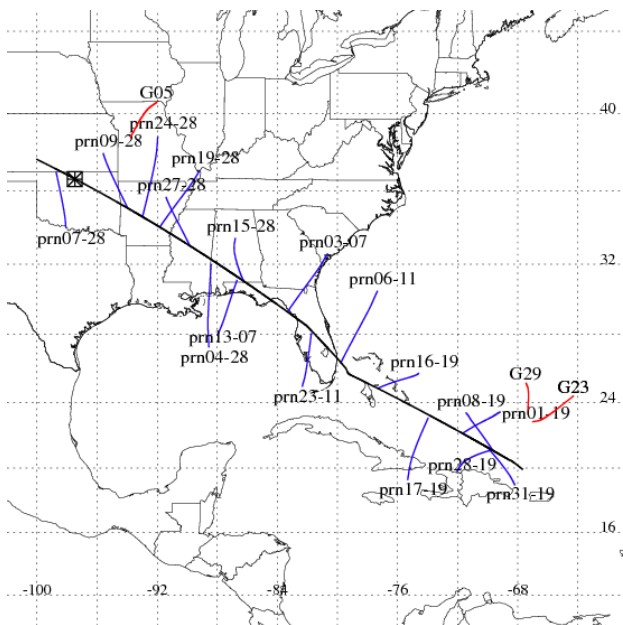

**Figure 1. Aircraft flight track (black) from the field station at St. Croix in the US Virgin Islands to the UCAR facility at Broomfield, Colorado on October 2, 2010. The tangent points (blue) and the PRNs of each ARO occultation are shown. The radiosonde from Lamont, Oklahoma (star) and the tangent points of the closest COSMIC RO soundings (red) to the flight track over the duration of the flight are also indicated.**





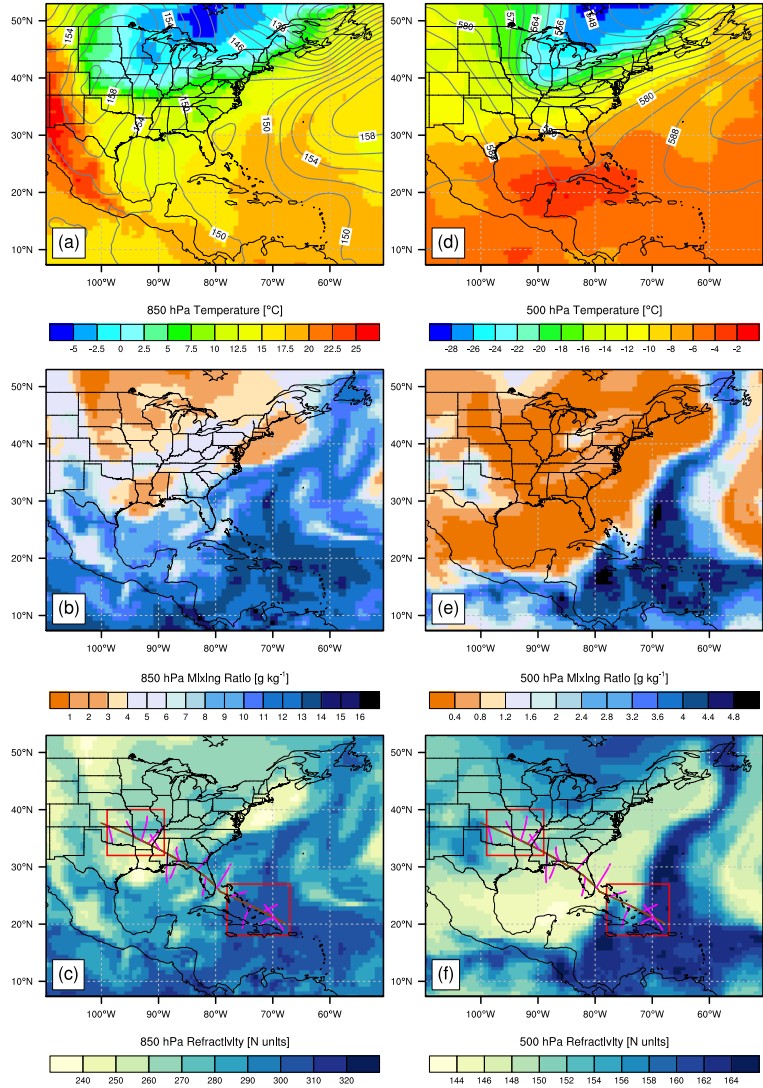

**Figure 2: (a, d) ERA-Interim temperature (K) overlaid with the geopotential height contour (dekametres, or tens of meters), (b, e) water vapor mixing ratio (g/kg), and (c, f) the derived refractivity (N-unit) at 850 hPa and 500 hPa, respectively, over the study region at 18Z on October 2, 2010. Also plotted in (c, f) are the flight track (brown), the ARO tangent points (purple) and the two boxes (red) indicating the selected regions with one over land (89–99° W, 32–40° N), and the other over the Caribbean sea (67–78° W, 18–27° N).**





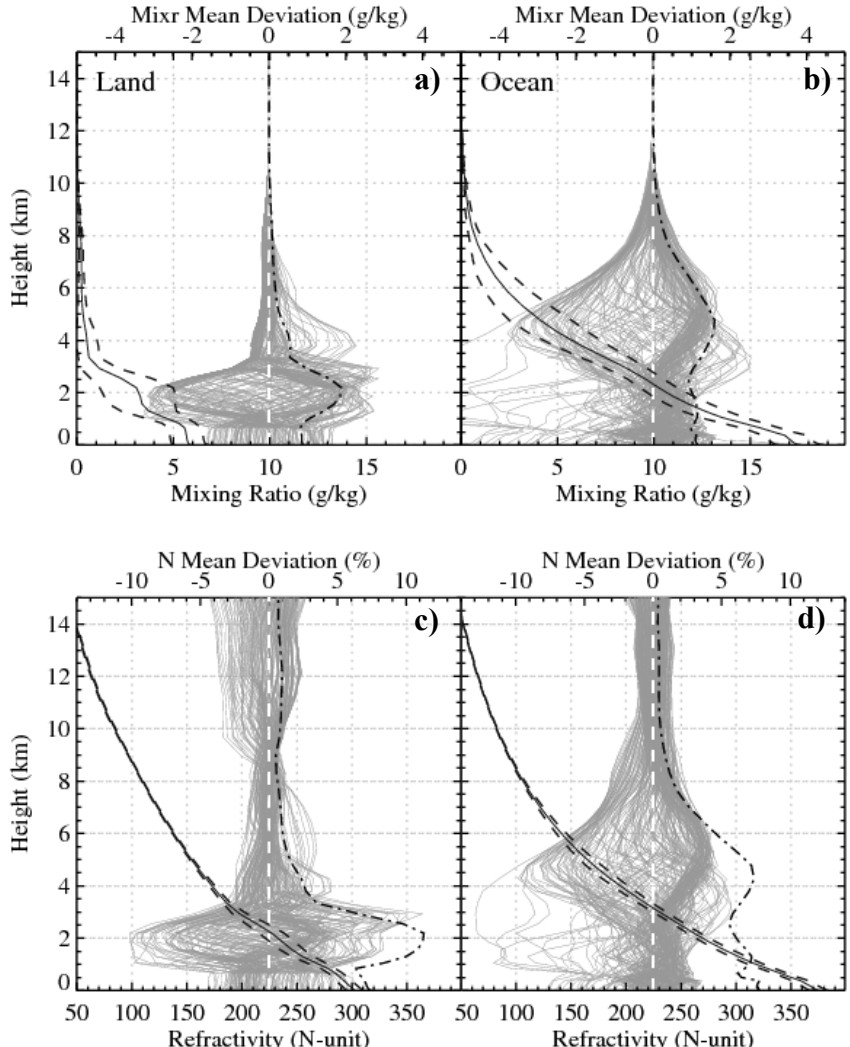

Figure 3: (a) ERA-I mean mixing ratio (solid black) and the mean ± standard-deviation (dashed) as well as the anomaly profiles (gray) and their respective standard deviation profiles (dash-dotted) over land, and (b) over ocean from the two selected regions indicated in Fig. 2c, at 18Z on October 2, 2010. (c) and (d) show the same but for ERA-I refractivity.



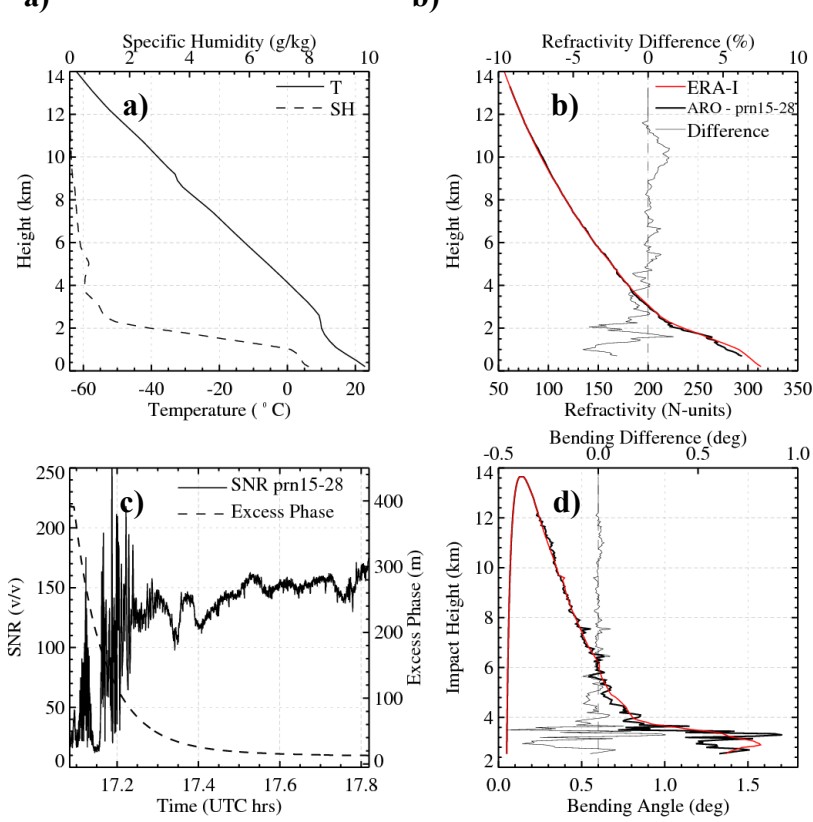

**Figure 4:** One typical ARO sounding (prn15-28) with the near-coincident ERA-I profiles, (a) ERA-I temperature and specific humidity profiles; (b) ARO (thick black) and the ERA-I (thick red) refractivity profiles along with their difference (thin black); (c) the SNR and excess phase of the ARO event; and (d) ARO (black) and the simulated ERA-I (red) bending angle profiles and their difference (thin black).





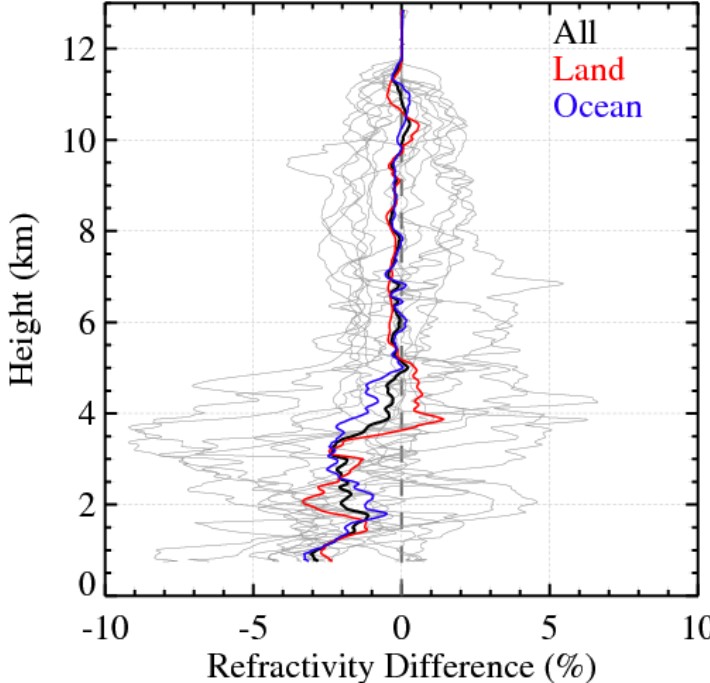

**Figure 5: Fractional refractivity difference between the ARO FSI retrievals and the near-coincident ERA-I profiles. Each thin gray line represents an individual occultation, and the three thick lines represent the mean difference for all profiles (black), and 5 the profiles over land (red) and over ocean (blue), respectively.**



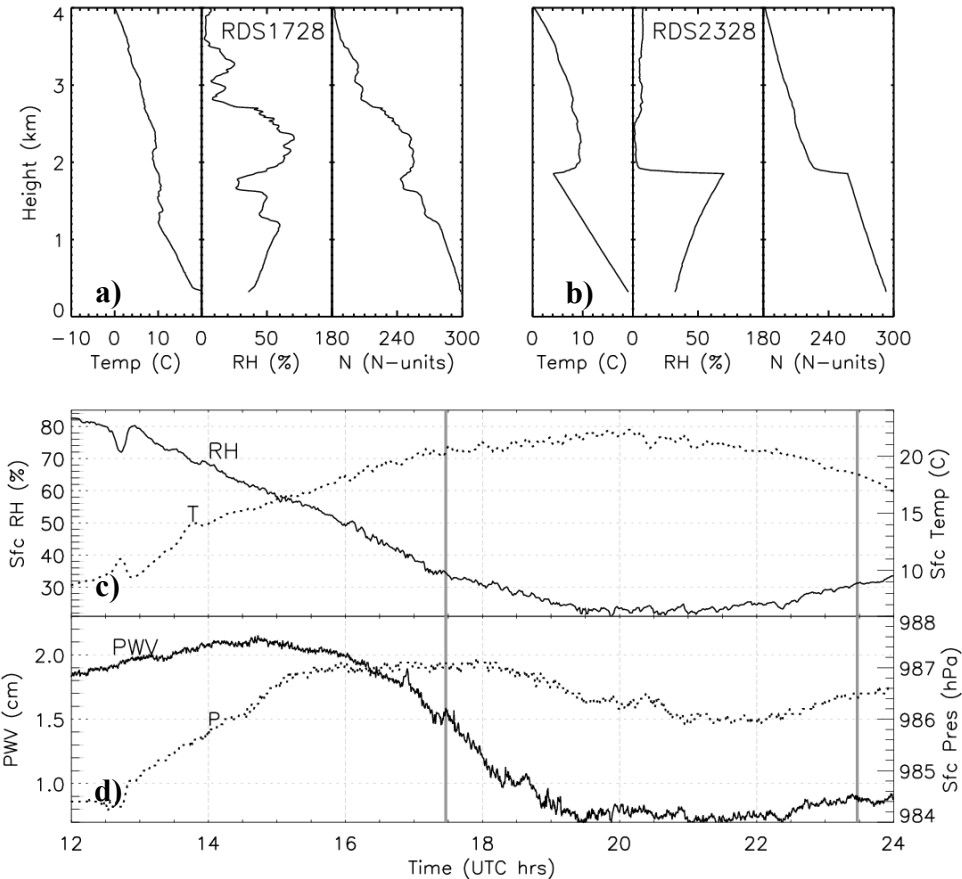

**Figure 6: (a, b) Vertical profiles of temperature, relative humidity and refractivity for radiosonde profiles at 17:28Z and 23:28Z, respectively; (c) time series of the surface relative humidity and temperature; and (d) surface pressure and precipitable water vapor (PWV) from the microwave radiometer, from the radiosonde station at Lamont, Oklahoma on October 2, 2010. The two vertical gray lines mark the time at 17:28Z and 23:28Z.**





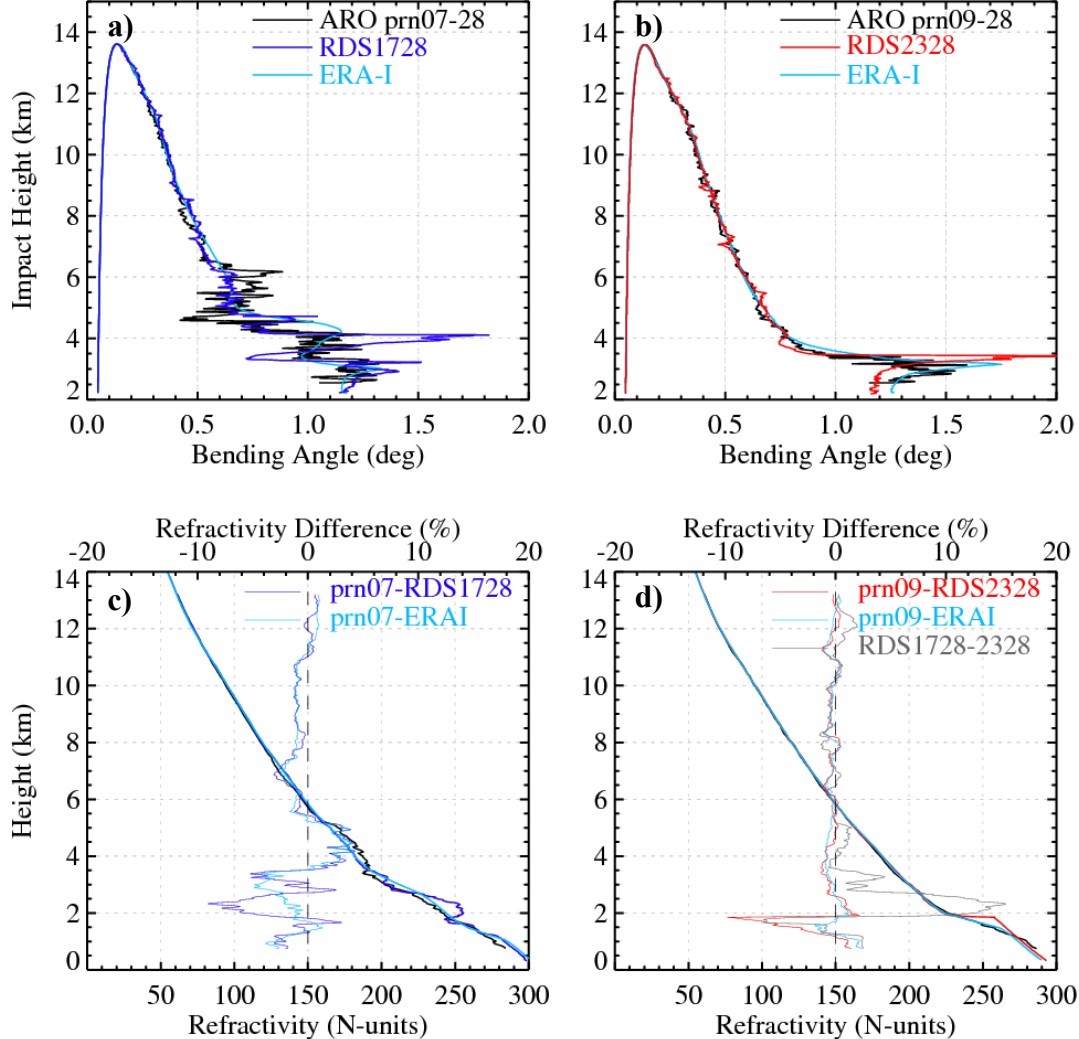

**Figure 7: ARO bending angle profiles for (a) prn07-28, and (b) prn09-28 along with the simulated bending angle of the near-coincident ERA-I and radiosonde profiles at 17:28Z and 23:28Z on October 2, 2010, respectively. (c, d) The refractivity profiles of ARO, radiosonde, and ERA-I and their difference including the fractional refractivity difference between the two radiosonde in (d).**



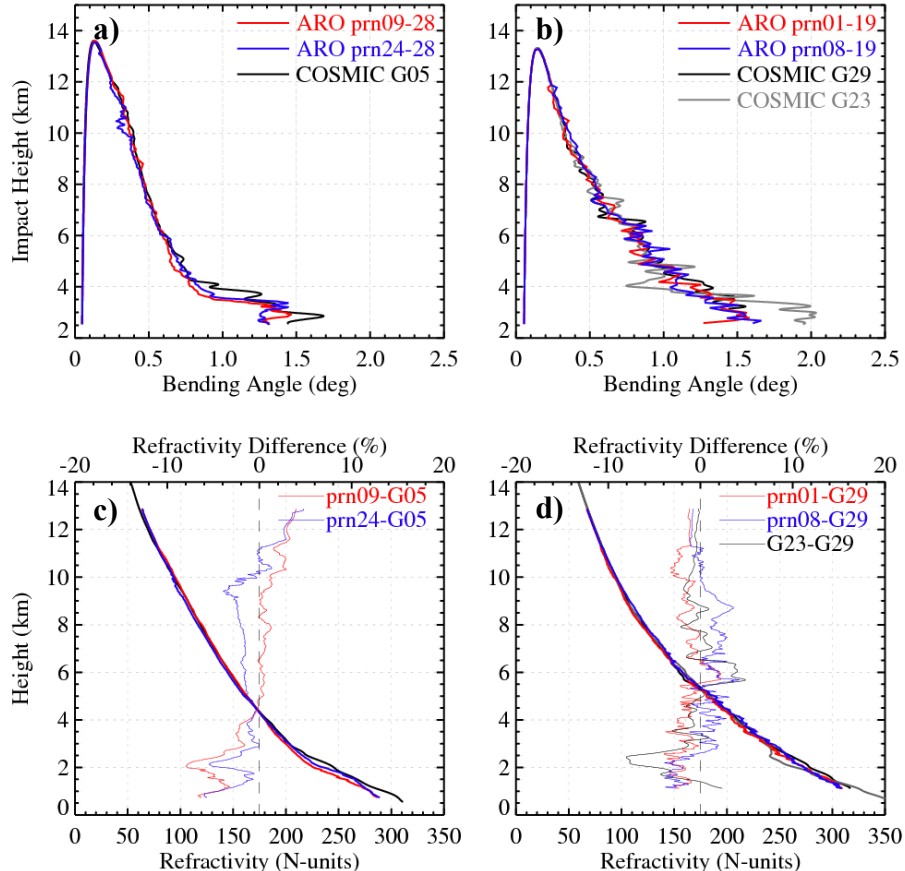

**Figure 8: (a) ARO bending angle for two ARO events, prn09-28 and prn24-28 along with the calculated bending of the closest COSMIC RO profile (G05); (b) ARO bending angle for two ARO events, prn01-19 prn08-19, along with the two closest COSMIC RO (G29, G23) profiles. (c, d) ARO and the COSMIC refractivity profiles, and fractional refractivity differences. The fractional refractivity difference between the two COSMIC profiles (G23, G29) is also shown in (d).**





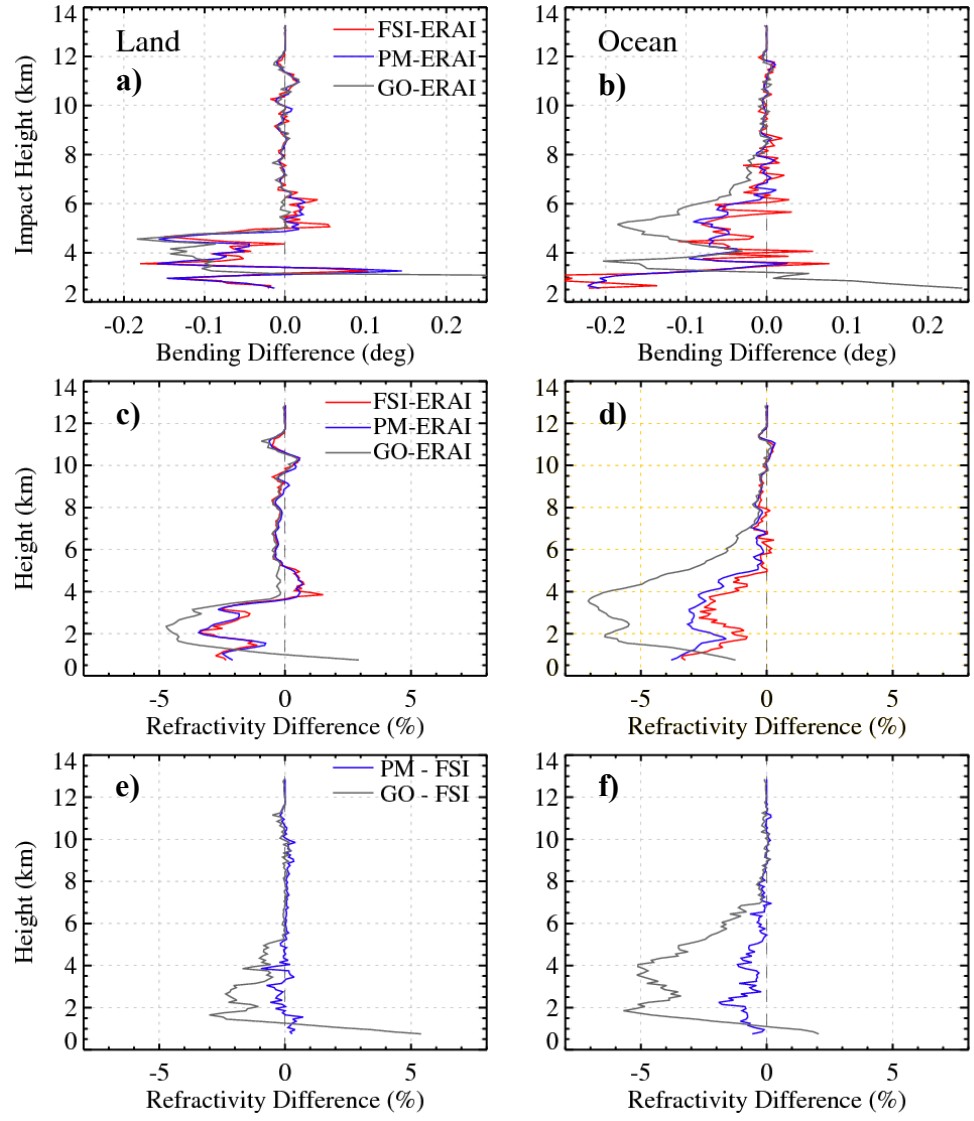

**Figure 9: (a, b) The difference between ARO bending angle from three retrieval methods (FSI, PM, and GO) and the simulated bending angle of near-coincident ERA-I profiles; (c, d) fractional refractivity difference between ARO retrievals and ERA-I profiles; (e, f) fractional refractivity difference of GO and PM retrievals from the FSI retrieval, over land and ocean, respectively.**





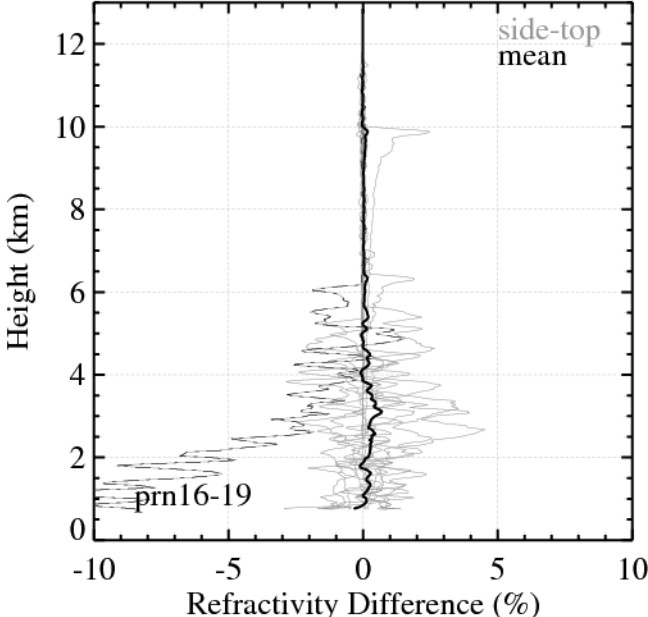

**Figure 10: The individual refractivity difference profiles (thin gray) between the ARO retrieval from side-looking antenna and the top antenna, which both track the same occulting GPS satellite. Also shown is the mean refractivity difference (thick black) that ignores the outlier case, prn16-19 (dashed) with large negative difference.**



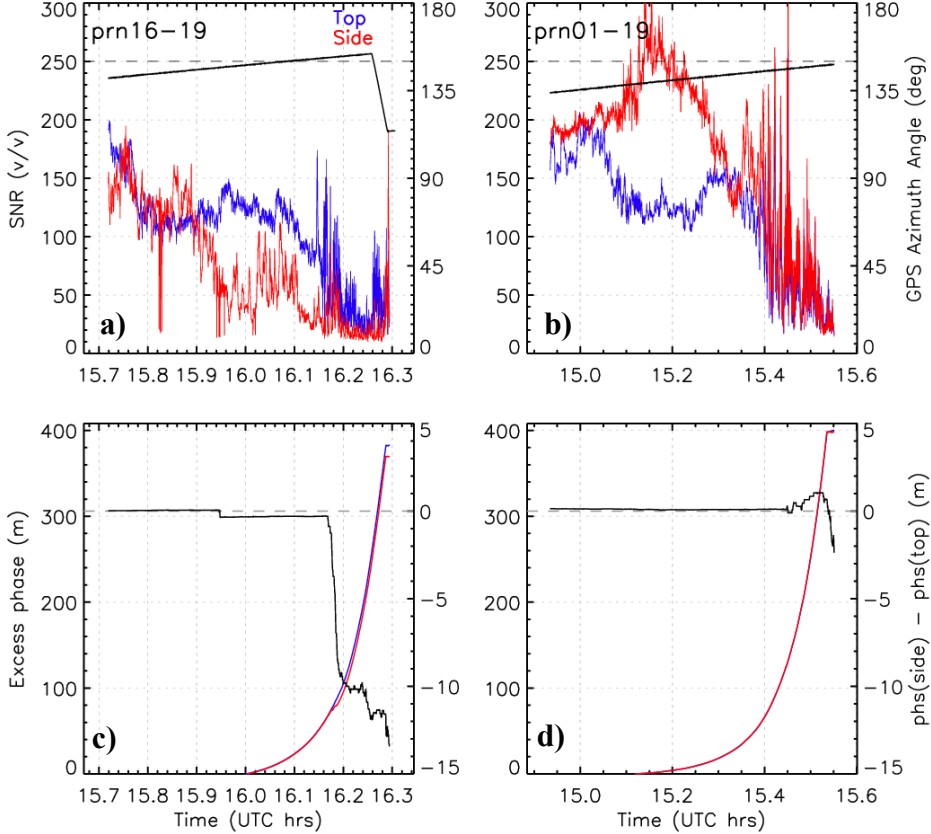

**Figure 11: The SNR and the azimuth angle relative to the flight heading direction for the occulting satellite (a) prn16 and (b) prn01, recorded from the top (blue) and the side-looking (starboard, red) antennas; the two horizontal dashed lines indicate the azimuth angle of 150°; (c, d) the excess phase for both antennas and their difference (black), and the two dashed lines indicates the zero phase difference.**