# Peer review of "Sensitivity of airborne radio occultation to tropospheric properties over ocean and land"

_Atmospheric Measurement Techniques, 2017_

## Referee Comment (RC1) · Anonymous Referee #1 · 14 Aug 2017

**Review: Sensitivity of airborne radio occultation to tropospheric properties over ocean and land**
**Xie et al.**

This manuscript presents the results of the ferry flight of an airborne campaign, during which different atmospheric conditions were found. The data were acquired for both a zenith-looking omnidirectional antenna and two limb-oriented ones and they have been processed with three techniques.

The topic and possibilities of the data set are relevant and worth publishing, but some aspects are not clear and require further work on the manuscript.

— The authors mention that Haase et al. 2014 and Murphy et al. 2015 did present results from analysis of RO data obtained during the same campaign. It is not very clear which are the main differences between this manuscript and the two former articles (except the use of the ferry flight instead of the scientific flights). Which are the new aspects tackled in this manuscript not covered in the former two?

— how the obtained numbers (precisions, biases, …) compare with the one obtained in Haase et al. 2014, Murphy et al., 2015 and any other paper on ARO analysis?

— I understood that these main differences could be either the open-loop vs. close-loop data or the comparison between different analysis (GO/FSI/PM) or the comparison between the different antennas. However, most of the presented work is based on antenna CH1, mostly FSI and only a short section is devoted to the comparison between different methods, antennas, etc. On the other hand, longer sections are devoted to compare the results with radio-sondes and COSMIC data, both data sets not optimal for the comparison (I would simply delete these sections, keep the comparison with ECMWF fields, and give more weight to the most novel aspects of the study).

— The abstract gives a lot of information about the final precision and biases with respect to the ancillary data sets, while, to my opinion, the novelty of the study lies on the diversity of hardware, low level processing and high level analysis (different antennas, open-loop vs. close-loop, GO/FSI/PM).

— The abstract seem to give two different numbers for RMS precision (0.3% in line 20, 0.2% in line 30)

— page 3, line 17: type PRECICT (PREDICT)

— page 4, line 10-15: the way the sentence is written, it seems that the use of 3 antennas was only done during this ferry flight, and not during the science flights of the same campaign. Is this interpretation correct? if so, this should be one of the main focuses of the manuscript, and the analysis, comparisons and conclusions derived from these special (thus unique) settings. If this is not true, then change the sentence to make it clearer.

— page 5, line 22: the authors do not comment on the potetential effects of setting the refractivity at the receiver level as given by ECMWF. How is this affecting the rest of the retrievals?

— page 6, lines 3-4: which are the effects of correcting the clock errors by pairing with another PRN? how would it change if using another PRN? Is this well-known and done before in RO or ARO? if so, please add references, otherwise comment on possible side effects.

— page 7, line 6, typo: bracket not close (PM, Wang et al. 2017

— page 8, line 6: the authors claim that the retrievals below 11.5 km are independent from the ECMWF (above 11.5 to the receiver level they set the refractivity to ECMWF fields). The fact that at a given range of altitudes they are fixed tot the ECMWF seems to me that it affects the rest of the retrievals, so they are not independent of the ECMWF fields. Could the authors comment on this?

— page 8, lines 14-17: new ERA-5 fields are now available, hourly at 25 km resolution. The interpolation process would be easier.

— page 10 and afterwards: the RMS difference with respect to ECMWF fields, are they computed up to 11.5 km or up to the receiver level? The comparison should stop at 11.5 km, as above this altitude the solution is set to ECMWF and including this upper segment in the RMS difference would artificially lower the number. Could the authors clarify this point? preferably in the body of the manuscript (not only in the rebuttal letter).

— pages 11 to 14: as mentioned before, I would remove the comparison with COSMIC data (and radiosondes), it is not optimal and does not contribute to the work, adds 'noise' to the study.

— page 15: the authors do not provide details about the antennas, could they present the gain of each of them? either the antenna or a pattern or something that might help understanding why an omnidirectional antenna looking to the zenith achieves ARO of better quality of the dedicated limb oriented ones. Perhaps Table 1 could include the gain of each of the antennas at the observation direction of each ARO event, or the azimuth direction w.r.t. the antenna boresight. Different colors in Figure 10 would also be a useful way to include this information.

— page 17, 3rd paragraph: rather useless comparisons, either remove or shorten, give less weight.
— Summary and conclusions: comparison with Haase et al., 2014 and Murphy et al., 2015 and other ARO experiments. Are these precisions and biases better or worse and why?

— page 18, line 10: it is remarkable that the precision with an omnidirectional antenna looking to the zenith achieves good or better precision than the dedicated side-looking antennas. This should be highlighted here. Also, authors could discuss the potential that this finding might have in the use of commercial flight GPS data for ARO (no need of dedicated side-looking antennas!).

— Figures 4, 5, 7, 8, 9 11, captions: which antenna? CH1? please add this information in the caption.

— Figures 4d, 7a, 7b, 8a, 8b: why the bending-impact profile starts at 2 km impact and climbs up to 14 and then down again? I don't understand the segment at low bending angle, with the impact parameter climbing up from 2 to 14 km. Is this well-known and understood?

— Figure 10: perhaps it would help to plot these line with a color code, accounting for either the gain of the antenna at the direction of the ARO event, or the azimuth difference w.r.t. the antenna boresight.

— Figure 11: why authors plot the 150 deg azimuth line? is this related to the orientation of the antenna boresight?

---

## Referee Comment (RC2) · Anonymous Referee #2 · 12 Sep 2017

General comments ————————

This is an involved paper, which examines various aspects of an Airborne Radio Occultation mission: the type of data, the nature of the processing, the antenna type and orientation, and the degree to which the results agree with various reference data sources. The resulting profiles compare favourably with those from reanalyses, and are worth publishing.

Specific comments ————————-

Why do the bending angle curves turn over at ∼13 km? Why are there two bending angles for each impact height below that? Please explain (in the text, not to me).

Technical corrections ————————

[Figure]

P8, L26: dimentional –> dimensional.

P9, L12: On the contrary –> In contrast.

All figures OK, except for the strange bending angle business.

———————————————

---

## Author Comment (AC1) · 5 Dec 2017

Xie et al., Sensitivity of airborne radio occultation to tropospheric properties over ocean and land

**We thank the editor and the two reviewers for the very insightful and constructive comments. A list of comments with detailed responses are shown below.**

**Response to the Anonymous Referee #1:**

1.  The authors mention that Haase et al. 2014 and Murphy et al. 2015 did present results from analysis of RO data obtained during the same campaign. It is not very clear which are the main differences between this manuscript and the two former articles (except the use of the ferry flight instead of the scientific flights). Which are the new aspects tackled in this manuscript not covered in the former two?

    **Yes, these recent papers, including Haase et al. (2014), Murphy et al. (2015) as well as Wang et al. (2016 and 2017), and the current paper, focused on the ARO measurements from the same PREDICT field campaign.**

    **Haase et al. (2014), Murphy et al. (2015) assessed the capabilities of the ARO observations specifically in the highly variable tropical hurricane environment from a conventional (close-loop tracking) receiver, so the results are limited to the upper part of the troposphere above 7 km, with only a preliminary description of open-loop tracking. Wang et al. (2016 and 2017) expand the error analysis of the tropical measurements to the lower part of the troposphere with the advantages of open-loop tracking (2016) and the phase matching inversion method (2017). These studies used only data from the top antenna because occulting satellite visibility to the side antennas was limited due to frequent turns during the PREDICT research flights.**

    **This paper offers several unique perspectives that are different from the previous three papers. (1) This paper focused on the ARO soundings from the ferry flight, which offers an optimal observing configuration (constant flight speed and altitude) for the side-looking antennas to collect high-quality ARO measurements that were not available during research flights. It evaluates whether on a straight flight, the high gain antennas present a significant advantage when combined with the previous recent advances. (2) In contrast to the studies in the hurricane environment (research flights), it investigates the ARO profile variability due to large scale synoptic variations in atmospheric boundary layer structure over land and ocean in a controlled experiment. (3) The simultaneous recording from two independent antennas allows the evaluation of ARO sounding quality and precision due to random errors, and confirms the high-quality ARO measurement from**

the relatively lower gain top antenna. (4) it compares three different retrieval methods, i.e., the full-spectrum-inversion (FSI), phase-matching (PM) and geometric-optics (GO), shedding light on the structural uncertainty of the ARO retrievals.

The "abstract" and the "Introduction" section have been updated to emphasize the unique aspects of this paper.

References:

Haase, J. S., B. J. Murphy, P.Muradyan, F. G. Nievinski, K. M. Larson, J. L. Garrison, and K.-N. Wang: First results from an airborne GPS radio occultation system for atmospheric profiling, Geophys. Res. Lett., 40, 1759–1765, doi:10.1002/2013GL058681, 2014.

Murphy, B., Haase, J. S., Muradyan, P., Garrison, J. L., and Wang, K.-N.: Airborne GPS radio occultation refractivity profiles observed in tropical storm environments, J. Geophys. Res., 120(5), 1690–1709, 2015.

Wang, K.-N., Garrison J. L., Acikoz U., Haase J. S., Murphy B. J., Muradyan P., Lulich T.: Open-loop tracking of rising and setting GPS radio-occultation signals from an airborne platform: Signal model and error analysis. IEEE Transactions on Geoscience and Remote Sensing. 54:3967-3984, 2016.

Wang, K.-N., Garrison J. L., Haase J. S., Murphy B. J.: Improvements to the GPS airborne radio occultation in the lower troposphere through implementation of the phase matching (PM) method, J. Geophys. Res., 122(10), 266–10,281. https://doi.org/10.1002/2017JD026568, 2017.

2. how the obtained numbers (precisions, biases, …) compare with the one obtained in Haase et al. 2014, Murphy et al., 2015 and any other paper on ARO analysis?

There are three previous papers analyzing the ARO sounding collected from the research flights during PREDICT campaign, which reflected three stages of advancement in the ARO sounding processing.

Both Haase et al. (2014) and Murphy et al. (2015) focused on the ARO soundings collected from the top antenna with the phase-locked loop tracking during the research flights. The GO retrieval method was used for both papers. *Murphy et al. (2015)* analyzed all ARO soundings collected from the research flight, and presented statistics on retrieval comparisons with model and radiosonde data in the upper and mid-troposphere. For example, the bias compared with ERA-I was -1% and standard deviation from 1.5% at 12 km to 2% at 7 km.

**Wang et al. (2017) did a comprehensive evaluation of ARO soundings (59 profiles from research flights) with open-loop (OL) tracking from the top antenna (CH1) with phase-matching retrieval that extends the ARO soundings from the aircraft altitude to near the surface. Significant improvement of ARO sounding quality in the lower troposphere was demonstrated compared to Haase et al. (2014) and Murphy et al. (2015), reducing the bias compared to ERA-I to near 0% from 4 to 12 km, and providing profiles from 2-4 km with a negative bias smaller than -1%. The standard deviation was 2% from 2 to 12 km.**

**The current paper analyzed the ARO profiles from the ferry flight with open-loop tracking. Both the FSI (this paper) and PM (Wang et al., 2017) retrieval methods show comparable retrieval quality in comparisons with ERA-I. They both show significant improvement in reducing bias in the moist lower troposphere (~1-2% below 4 km) compared to the GO method (3-4% below 4 km) as seen in Figure 9 (c-f) from this paper and Figure 8 from Wang et al. (2017) below.**

[Figure]

**Figure: The fractional difference of the refractivity retrievals (ARO minus ERA-I) for all 59 profiles from the 5 research flights retrieved by (a) GO and (b) PM are shown. For the PM results, the bias with respect to ERA-I, remains less than 1% and the standard deviation is less than 2% over the entire range above 2 km. (Adapted from Figure 8 in Wang et al., 2017)**

**However, it is interesting to note that in this environment where sharp layer (including the boundary layer) structure in the lower troposphere was much more developed than in the hurricane flights, the tendency for ERA-I to produce a vertically smoother approximation over land (Fig. 6, Fig. 8c,d) resulted in a systematic positive *N*-bias at around 4 km and a negative bias below that on the order of 2-3 %.**

3.  I understood that these main differences could be either the open-loop vs. close-loop data or the comparison between different analysis (GO/FSI/PM) or the comparison between the different antennas. However, most of the presented work is based on antenna CH1, mostly FSI and only a short section is devoted to the comparison between different methods,

antennas, etc. On the other hand, longer sections are devoted to compare the results with radiosondes and COSMIC data, both data sets not optimal for the comparison (I would simply delete these sections, keep the comparison with ECMWF fields, and give more weight to the most novel aspects of the study).

**We deleted the Section 4.3 & Figure 8 that compares the ARO with COSMIC soundings.**

**However, we kept the section comparing the ARO with the radiosonde data, as the two radiosonde profiles used for the study shows a very different vertical atmospheric structure before and after the frontal passage. The analysis demonstrated that ARO soundings captured the major change of the fine vertical structure at the top and inside the PBL.**

**Even though it is only a case study, we believe it is important to highlight this sensitivity of the ARO soundings to the fine vertical structure shown in the collocated radiosonde profile.**

4. The abstract gives a lot of information about the final precision and biases with respect to the ancillary data sets, while, to my opinion, the novelty of the study lies on the diversity of hardware, low level processing and high level analysis (different antennas, open-loop vs. close-loop, GO/FSI/PM).

   **We have updated the abstract to emphasize on the unique contribution of the paper, which is described in more detail in the response-to-comment #1.**

5. The abstract seem to give two different numbers for RMS precision (0.3% in line 20, 0.2% in line 30)

   **The two numbers are different. The 0.3% is the RMS difference between ARO and the collocated ERA-I profiles, which could be considered an indicator of the accuracy of ARO measurement, assuming the ERA-I is a good, independent approximation of the true atmosphere. On the other hand, the 0.2% is the ARO refractivity retrieval difference between the top (CH1) and the side (CH2, CH3) antennas, which is a good indicator of the precision of the ARO measurement.**

6. page 3, line 17: type PRECICT (PREDICT)

   **Corrected, thanks.**

7. page 4, line 10-15: the way the sentence is written, it seems that the use of 3 antennas was only done during this ferry flight, and not during the science flights of the same campaign. Is this interpretation correct? if so, this should be one of the main focuses of the manuscript, and the analysis, comparisons and conclusions derived from these special (thus unique) settings. If this is not true, then change the sentence to make it clearer.

**All three antennas collected ARO measurements during both the research flights and the ferry flight, however, the ARO measurement from side antennas (CH2, CH3) during the research flights produced significantly fewer and degraded quality profiles compared to the top antenna due to aircraft turns. Therefore, only ARO measurements collected from the top antenna were presented in Haase et al. (2014), Murphy et al. (2015) and Wang et al. (2017).**

**And the reviewer is absolutely correct, one of the major contribution of this paper is to take advantage of the ARO measurements from the high-gain side antenna (CH2, CH3) during the ferry flight to evaluate the ARO soundings collected from the lower gain zenith antenna (CH1).**

**The manuscript has been modified to clarify the statement as follows:**

**Page 4, line 6-10,**
*The ARO signals were simultaneously recorded from two high-gain antennas mounted on both sides of the aircraft fuselage, and one lower gain zenith antenna on the top of the aircraft (Haase et al., 2014, Murphy et al., 2015; Wang et al., 2016). However, the complicated research flight patterns, i.e., sometimes changing flight direction during an occultation, led to degraded signal tracking from the side-looking antennas as the line-of-sight deviated from the maximum in the GPS antenna gain pattern (Murphy et al., 2015; Wang et al., 2016).*

**Page 4, line 14-17,**
*"This long flight at nearly constant altitude (~13 km) provided an optimal configuration for simultaneous high-quality ARO measurements from two side-looking antennas and one top antenna. Such independent recordings of occultation events from multiple antennas allows the evaluation of ARO sounding quality and precision."*

**Page 1 line 31 (Abstract)**
*"The unique opportunity to make simultaneous independent recordings of occultation events from multiple antennas establishes that high precision ARO measurements can be achieved corresponding to an RMS difference better than 0.2 % in refractivity (or ~0.4 K)."*

8. page 5, line 22: the authors do not comment on the potential effects of setting the refractivity at the receiver level as given by ECMWF. How is this affecting the rest of the retrievals?

**The impact of any error in the assumed refractivity at the receiver on the ARO retrieval has been evaluated in detail in Xie et al. (2008). The simulation study demonstrated that the random error of refractivity of 0.5% (~ 1 K) at the receiver results in errors up to 0.2% refractivity close to the aircraft altitude, and decreases quickly to 0.05% at 100 m below the receiver height. This error is much smaller than the retrieval errors due to the small**

**accumulated excess Doppler relative to the random Doppler noise, which produces amplified bending angle errors near (within about 1 km) of the aircraft altitude.**

**Accounting for refractivity at the receiver location has been implemented in the ARO processing in Murphy et al., 2015 (Fig. 4) and Wang et al., 2017, using the flight level in-situ measurements, and also in Adhikari et al. (2016) and this paper using interpolated ERA-I estimates of refractivity. This produces a closer agreement between the ARO and ERA-I profiles at flight level in the latter two studies, as would be expected.**

**In this paper, to mitigate the high bending angle errors near the aircraft altitude, the bending angle within the top 1.5 km is replaced with the simulated bending angle profiles obtained from collocated ERA-I profiles. Only refractivity retrievals below ~11.5 km are interpreted. The impact of this assumption decays exponentially below that height (Adhikari et al., 2016).**

9. page 6, lines 3-4: which are the effects of correcting the clock errors by pairing with another PRN? how would it change if using another PRN? Is this well-known and done before in RO or ARO? if so, please add references, otherwise comment on possible side effects.

   **The ARO receiver is tracking the occulting GPS satellite at low elevation and multiple high-elevation GPS satellites simultaneously during an occultation (e.g., Wang et al., 2016). Murphy (unpublished) compared different satellite pairs and found that using the high elevation satellite with the highest SNR produced the best results. It is a standard procedure for space RO, e.g., COSMIC CDAAC, to publish occultations for multiple reference satellites (http://cdaac-www.cosmic.ucar.edu).**

   **Reference:**

   **Wang, K.-N., Garrison J. L., Acikoz U., Haase J. S., Murphy B. J., Muradyan P., Lulich T.: Open-loop tracking of rising and setting GPS radio-occultation signals from an airborne platform: Signal model and error analysis. *IEEE Transactions on Geoscience and Remote Sensing*. 54:3967-3984, 2016.**

10. page 7, line 6, typo: bracket not close (PM, Wang et al. 2017

    **Corrected, thanks.**

11. page 8, line 6: the authors claim that the retrievals below 11.5 km are independent from the ECMWF (above 11.5 to the receiver level they set the refractivity to ECMWF fields). The fact that at a given range of altitudes they are fixed to the ECMWF seems to me that it affects the rest of the retrievals, so they are not independent of the ECMWF fields. Could the authors comment on this?

**We inaccurately described the error mitigation, which has been discussed in detail in response-to-comment #8.**

**We have removed the following line:**
"*As a result, the independent refractivity retrievals from ARO measurements extends from surface up to about 11.5 km, which is ~1.5 km below the receiver altitude.*"

**And replaced it with the following sentence:**
L28-pp8: "*The dependence on any error in the ERA-I refractivity assigned to the receiver altitude decays exponentially as height decreases.*"

12. page 8, lines 14-17: new ERA-5 fields are now available, hourly at 25 km resolution. The interpolation process would be easier.

    **Yes, the higher temporal and spatial resolution from ERA-5 could lead to a more precise interpolation of the refractivity values at the tangent point locations. However, the limitations of comparing a model point value with the ARO tangent point value derived from a horizontal integral of the atmospheric properties would still remain, and is responsible for larger standard deviation of differences at lower altitudes.**

    **The ERA-5 was not available at the time of submission of the paper nor at the time when the retrievals and most of the analysis was done. Future work may use the ERA-5 fields for comparisons, although we would use the same interpolation process, per se. ECMWF is currently evaluating ERA-5 biases relative to spaceborne RO (Sean Healy, personal communication) so we are not yet planning to use it as a standard.**

13. page 10 and afterwards: the RMS difference with respect to ECMWF fields, are they computed up to 11.5 km or up to the receiver level? The comparison should stop at 11.5 km, as above this altitude the solution is set to ECMWF and including this upper segment in the RMS difference would artificially lower the number. Could the authors clarify this point? preferably in the body of the manuscript (not only in the rebuttal letter).

    **We have updated all the statistics so they are calculated only up to 1.5 km below the aircraft. The aircraft altitude changes slightly during the ferry flight, so the height up to which the statistics are calculated varies from 11.5 to 11.9 km. All the numbers either remain the same or change less than 0.1 %. The numbers have been updated in the manuscript.**

14. pages 11 to 14: as mentioned before, I would remove the comparison with COSMIC data (and radiosondes), it is not optimal and does not contribute to the work, adds 'noise' to the study.

    **We deleted the Section 4.3 & Figure 8 that compares the ARO with COSMIC soundings.**

**However, we kept the Section 4.2 that compares the ARO with the radiosonde data, as the two radiosonde profiles used for the study shows a very different vertical atmospheric structure before and after the frontal passage. The analysis demonstrated that ARO soundings captured the major change of the fine vertical structure inside the PBL.**

**Even though it is only a case study, we believe it is important to highlight this sensitivity of the  ARO soundings to the fine vertical structure shown in the collocated radiosonde profile.**

15. page 15: the authors do not provide details about the antennas, could they present the gain of each of them? either the antenna or a pattern or something that might help understanding why an omnidirectional antenna looking to the zenith achieves ARO of better quality of the dedicated limb oriented ones. Perhaps Table 1 could include the gain of each of the antennas at the observation direction of each ARO event, or the azimuth direction w.r.t. the antenna boresight. Different colors in Figure 10 would also be a useful way to include this information.

    **We have added one figure (new Figure 1) to illustrate the antenna gain pattern for the high-gain, narrow vertical side-looking (port) antenna (starboard is similar to the port).**

[Figure]

Figure 1. (a) Approximate location of three antennas on HIAPER aircraft, (b, c) the port side-looking antenna (CH2) azimuthal and elevation gain pattern, respectively. For the port antenna,  0° azimuth and elevation are oriented toward the horizon perpendicular

to the flight direction, 90° in azimuth points forward and 90° in elevation points at nadir direction. Note that (b, c) are adapted from Figure 18 in Wang et al., (2016).

**Note the ARO signals are focused at about ±5° elevation angle, with nearly uniform gain over that range for the port antenna (Fig. 1c). However, the azimuth relative to the boresight varies significantly among occultations. Unlike the omnidirectional top antenna with an isotropic azimuthal gain pattern (not shown), the side-looking antennas have a drop of 10 dB from the boresight to +/- 40° azimuth. Such antenna directivity, and nulls near -60° in the gain pattern, are the most likely explanation for the lower quality results (Fig. 10) for the side-looking antennas than the top antenna, which has the advantage of wide-view capability.**

**The side-looking antenna is not expected to record occulting GPS at large azimuth angles, especially over -60° (towards the rear of the aircraft). However, several successful recordings with azimuth angle as large as 77.4° (towards the front of the aircraft) were collected. We did not find a clear correlation between the quality of the ARO sounding to the azimuth angle when successful recording is achieved, and therefore did not differentiate ARO profiles by azimuth angle in Fig. 10.**

**However, as suggested, a new column is added to show the azimuth angle relative to boresight of the lowest tangent point for each ARO event in Table 1. Note the four shaded ARO events that missed the side-looking antenna recording are due to the large azimuth angle away from the antenna boresight.**

16. page 17, 3rd paragraph: rather useless comparisons, either remove or shorten, give less weight.

   **The paragraph has been significantly shortened as follows:**

   **L16-pp17:** "*The ARO soundings also agree well with the near-coincident radiosonde above 4 km and capture the heights of sharp layers in the PBL and the variations observed by the radiosondes during the cold frontal passage (Fig. 8).*"

   **It is interesting to note that in this environment where sharp layer (including the boundary layer) structure in the lower troposphere was much more developed than in the hurricane flights, the tendency for ERA-I to produce a vertically smoother approximation over land (Fig. 6, Fig. 8c,d, Fig. 9c) resulted in a systematic positive bias at around 4 km and a negative bias below that on the order of 2-3%. In addition, the difference in PBL height at the location of the radiosonde and the ARO tangent point can produce large positive and negative differences in refractivity.**

17. Summary and conclusions: comparison with Haase et al., 2014 and Murphy et al., 2015 and other ARO experiments. Are these precisions and biases better or worse and why?

    **As detailed discussed in the response-to-comment #1, the "accuracy" of ARO sounding as compared with the ERA-I reanalysis from this paper is improved relative to the geometric optics inversion method employed on the phase-locked loop data in Haase et al. 2014, and Murphy et al., 2015. The precision and bias of the FSI and PM is comparable to that in Wang et al., 2017. The differences between ERA-I and the ARO profiles are bigger in the lowest troposphere here than in Wang et al., 2017 because of the differences in the precisely retrieved boundary layer height, particularly over land (Fig. 9c), compared to the previous study in the tropical environment.**

18. page 18, line 10: it is remarkable that the precision with an omnidirectional antenna looking to the zenith achieves good or better precision than the dedicated side-looking antennas. This should be highlighted here. Also, authors could discuss the potential that this finding might have in the use of commercial flight GPS data for ARO (no need of dedicated side-looking antennas!).

    **Yes, we agree with the reviewer here. The following sentence is added at the end of the Section 5.**

    **"***The surprisingly good quality of ARO measurements from a simple omnidirectional zenith antenna greatly simplifies the implementation of the ARO system and increases the feasibility of developing an operational tropospheric sounding system on-board commercial aircrafts in the future, which could provide a large amount of data for direct assimilation in numerical weather prediction models.***"**

19. Figures 4, 5, 7, 8, 9 11, captions: which antenna? CH1? please add this information in the caption.

    **All the ARO profiles from Fig. 5, 6, 8, 9 are from top antenna (CH1). The "CH1" has been added into the caption. CH1 and CH3 were added into the caption of Fig. 10, 11.**

20. Figures 4d, 7a, 7b, 8a, 8b: why the bending-impact profile starts at 2 km impact and climbs up to 14 and then down again? I don't understand the segment at low bending angle, with the impact parameter climbing up from 2 to 14 km. Is this well-known and understood?

    **The impact height is the difference between impact parameter and the radius of the Earth, whereas the impact parameter is the product of refractive index and radius of the curvature at the tangent point (i.e., impact height = ($n \cdot R$) - Re). The impact parameter is widely used in the radio occultation community as it is the constant quantity for each single ray path connecting the GPS and the RO receiver. The bending angle retrieval is**

normally in terms of the impact parameter instead of the geometric height. After the "Abel inversion", the refractivity is then retrieved at the geometric height (e.g, Kursinski et al., 1997).

With the surface refractivity close to 350 (*N*-units), the impact height near the surface is close to ~2 km. At the aircraft altitude (~13 km), the impact height (~13.5 km) becomes much closer to the geometric height due to the exponential decrease of refractive index of the atmosphere.

Reference:
Kursinski, E. R., G. A. Hajj, J. T. Schofield, R. P. Linfield, and K. R. Hardy (1997), Observing Earth's atmosphere with radio occultation measurements using the Global Positioning System, *J. Geophys. Res.*, 102(D19), 23429–23465, doi:10.1029/97JD01569.

The segment of increasing bending angle is corresponding to the positive elevation bending angle measurements that is well-known for airborne RO (e.g., Healy et al., 2002, Xie et al., 2008). The bending angle increases as the elevation angle decreases from about +5° to 0° (at the horizon), when RO signals go through thicker atmosphere.

21. Figure 10: perhaps it would help to plot these line with a color code, accounting for either the gain of the antenna at the direction of the ARO event, or the azimuth difference w.r.t. the antenna boresight.

The side-looking antenna is not expected to record occulting GPS at large azimuth angles, especially over -60° (towards the rear of the aircraft). However, several successful recordings with azimuth angle as large as 77.4° (towards the front of the aircraft) were collected. We did not find a clear correlation between the quality of the ARO sounding to the azimuth angle when successful recording is achieved, and therefore did not differentiate ARO profiles by azimuth angle in Fig. 10.

However, as suggested, a new column is added to show the azimuth angle relative to boresight of the lowest tangent point for each ARO event in Table 1. Note the four shaded ARO events that missed the side-looking antenna recording are due to the large azimuth angle away from the antenna boresight.

22. Figure 11: why authors plot the 150 deg azimuth line? is this related to the orientation of the antenna boresight?

The azimuth angle was reference to the flight heading direction. To avoid confusion, we have updated the azimuth angle to be referenced to the antenna boresight direction (i.e., perpendicular to the flight direction in the horizon) in Figure 11 with negative sign indicating towards the rear of the aircraft. The azimuth angles at lower tangent heights are over −60°, where the near null antenna gain (new Fig. 1b) results in low SNR and degraded ARO sounding quality.

---

## Author Comment (AC2) · 5 Dec 2017

Xie et al., Sensitivity of airborne radio occultation to tropospheric properties over ocean and land

**We thank the editor and the two reviewers for the very insightful and constructive comments. A list of comments with detailed responses are shown below.**

**Response to the Anonymous Referee #2:**

General comments
This is an involved paper, which examines various aspects of an Airborne Radio Occultation mission: the type of data, the nature of the processing, the antenna type and orientation, and the degree to which the results agree with various reference data sources. The resulting profiles compare favourably with those from reanalyses, and are worth publishing.

**We thank the reviewer's very positive comments.**

Specific comments

1. Why do the bending angle curves turn over at 13 km? Why are there two bending angles for each impact height below that? Please explain (in the text, not to me).

   **Thanks for the suggestion. We have added the following paragraph into Section 4.1 (ARO retrievals with near-coincident ERA-I profiles) to further discuss the unique aspect of the ARO bending angle measurements.**

   **L13-27, pp11:** "*Note the ERA-I bending angle profile is simulated based on the modified forward Abel integration of the refractivity (e.g., Xie et al., 2008). For an ARO receiver located inside the atmosphere, the GNSS signals from both the positive and negative elevation angle (typically $\pm 5°$ reference to the local horizon) are recorded to retrieve the bending angles from the surface up to the receiver altitude. Assuming a spherically symmetric atmosphere, for every negative elevation ray bending angle, there is a corresponding positive elevation bending angle with the same impact parameter. The partial bending angle, i.e., the difference between the negative and positive elevation bending angle, can then be derived and converted to refractivity through a modified inverse Abel transformation (Healy et al., 2002; Lesne et al., 2002; Xie et al., 2008). For illustration purposes in Figure 5, the impact height is used, which is simply the difference between the impact parameter and the local curvature radius of the Earth. Because impact height depends on refractivity, it is typically a value of about 2 km at the surface in the tropics. Note the simulated bending angles from positive elevation angles (e.g., close to +5°) are*

*generally very small, because the GNSS signals go through the relatively dry and low density atmosphere above the aircraft altitude. The bending angles increase up to ~0.15° (Fig. 5d) at zero elevation (at the local horizon), when the tangent point of the ray is at the aircraft location at ~13.5 km altitude (corresponding to the maximum impact height of ~14 km, in Fig. 5d). The bending angles continue to increase at lower negative elevation angle as the GPS signals go through the denser and moister atmosphere."*

Technical corrections

1. P8, L26: dimentional –> dimensional.

   **Corrected, thanks.**

2. P9, L12: On the contrary –> In contrast.

   **Corrected, thanks.**

3. All figures OK, except for the strange bending angle business.

   **Corrected, thanks.**